# Storms drive outgassing of $CO_2$ in the subpolar Southern Ocean

Sarah-Anne Nicholson [1✉], Daniel B. Whitt [2,3], Ilker Fer [4], Marcel D. du Plessis [1,5,6], Alice D. Lebéhot[1,6,7], Sebastiaan Swart[5,6], Adrienne J. Sutton [8] & Pedro M. S. Monteiro [1,6]

The subpolar Southern Ocean is a critical region where $CO_2$ outgassing influences the global mean air-sea $CO_2$ flux ($F_{CO2}$). However, the processes controlling the outgassing remain elusive. We show, using a multi-glider dataset combining $F_{CO2}$ and ocean turbulence, that the air-sea gradient of $CO_2$ ($\Delta pCO_2$) is modulated by synoptic storm-driven ocean variability (20 µatm, 1–10 days) through two processes. Ekman transport explains 60% of the variability, and entrainment drives strong episodic $CO_2$ outgassing events of 2–4 mol m$^{-2}$ yr$^{-1}$. Extrapolation across the subpolar Southern Ocean using a process model shows how ocean fronts spatially modulate synoptic variability in $\Delta pCO_2$ (6 µatm$^2$ average) and how spatial variations in stratification influence synoptic entrainment of deeper carbon into the mixed layer (3.5 mol m$^{-2}$ yr$^{-1}$ average). These results not only constrain aliased-driven uncertainties in $F_{CO2}$ but also the effects of synoptic variability on slower seasonal or longer ocean physics-carbon dynamics.

[1] Southern Ocean Carbon-Climate Observatory (SOCCO), CSIR, Cape Town, South Africa. [2] National Center for Atmospheric Research, Boulder, CO, USA. [3] NASA Ames Research Center, Mountain View, CA, USA. [4] Geophysical Institute, University of Bergen, Bergen, Norway. [5] Department of Marine Sciences, University of Gothenburg, Gothenburg, Sweden. [6] Department of Oceanography, University of Cape Town, Cape Town, South Africa. [7] Marine and Antarctic Research centre for Innovation and Sustainability (MARIS), University of Cape Town, Cape Town, South Africa. [8] NOAA Pacific Marine Environmental Laboratory, Seattle, WA, USA. ✉email: snicholson@csir.co.za

The Southern Ocean is a key component of the Earth's carbon budget. It accounts for 40–50% of the total mean annual ocean uptake of anthropogenic $CO_2$ (~1 Pg C yr$^{-1}$)[1–4]. In addition, an increase of the annual mean outgassing of natural $CO_2$ from the Southern Ocean in the 1990s and the subsequent decrease in outgassing in the 2000s (resulting in a ~0.5 Pg C yr$^{-1}$ reinvigoration of ocean uptake) showed that the global ocean carbon budget is sensitive to variability in the Southern Ocean[5–10]. This outgassing variability has been linked to climate-mode forced variations in wind-driven upwelling that comprises the surfacing of deep waters with high concentrations in dissolved inorganic carbon (DIC) resulting in widespread but variable outgassing of $CO_2$ in the subpolar region, which counteracts the $CO_2$ uptake flux[2,9,10]. However, it is not well understood what role the daily-to-seasonal physics of the surface mixed layer, the critical boundary between the atmosphere and the upwelled reservoir of DIC, plays in modulating the magnitude of this outgassing flux[11,12].

The subpolar outgassing region, south of the Polar Front, coincides with the core of Southern Hemisphere storm tracks (~50–65°S)[13,14] and is regarded as the windiest region in the world. Strong storms occur in regular succession (4–8 days[15]) throughout the year, inflicting intense (>0.8 N m$^{-2}$ [16]) but short-lived (2–3 days[15]) surface wind stress. In the mid-latitudes of both hemispheres, storms energize the mixed layer at synoptic timescales (1–10 days), triggering enhanced vertical mixing (entrainment) and advection that result in surface and subsurface ocean exchanges of heat, momentum and chemical properties. Thus, storms drive significant variability in biogeochemistry[11,17–20] and the amplitude and timing of seasonal mixed-layer depths[21,22]. Yet, despite these well-documented upper-ocean responses, there remains limited understanding of what impact these frequently passing storms have individually or cumulatively on the outgassing magnitude and variability in this globally critical region, the subpolar Southern Ocean.

The lack of understanding of how storms impact this region, may be further reflected by the strong biases in the seasonal cycle of ocean $CO_2$ in both Earth System Models and forced ocean models in the Southern Ocean, which exclude sub-grid scale mixed-layer dynamics[23–25]. Model experiments suggest that the exclusion of processes associated with storms (e.g. near-inertial waves) significantly reduces vertical mixing and the surface ocean supply of DIC, reducing winter outgassing and increasing summer uptake of $CO_2$[11]. Moreover, recent high-resolution (10-day) observations from biogeochemical floats have revealed an underestimation of the magnitude of $CO_2$ outgassing in ship-based contemporary estimates due to significant spatial-temporal aliasing by these observations[26,27]. Autonomous surface vehicle sampling at higher resolution (hourly or faster) shows further temporal aliasing of floats (i.e. a 10-day sampling interval may result in ~20% uncertainty in the mean $CO_2$ flux). This aliasing is likely a result of unresolved mixed-layer responses to highly variable synoptic events[17,28]. There remains a significant gap in the understanding of the mechanisms that drive variability on synoptic timescales and how this synoptic variability rectifies on the seasonal cycle and mean of $CO_2$ fluxes.

Here, we address these gaps through high-resolution atmosphere-ocean observations from autonomous vehicles in the Atlantic sector of the subpolar Southern Ocean. We find that storms modulate the direction and magnitude of the air-sea $CO_2$ gradient ($\Delta pCO_2$) and flux ($F_{CO2}$) through two mixed-layer processes. Synoptic variability in the wind-driven Ekman flow advects upwelled DIC-rich waters from the south (and low DIC waters from the north) and explains most of the $\Delta pCO_2$ variability on timescales from a few days to months. In addition, intense wind events drive turbulent entrainment of the deeper DIC reservoir into the surface mixed layer and strong outgassing $F_{CO2}$. We construct a process model, which captures the observed

$\Delta pCO_2$ variability to estimate its relevance across the subpolar Southern Ocean and on longer timescales. We show that Ekman-driven synoptic variance of $\Delta pCO_2$ is spatially modulated by large-scale ocean fronts, where meridional DIC gradients are strong hence Ekman flows drive large DIC transports. However, Ekman-driven synoptic variance of $\Delta pCO_2$ is largely oscillatory with little additive effect on the mixed-layer DIC budget on timescales longer than a couple of months. On the other hand, storm-driven entrainment, which is spatially modulated by variations in seasonal stratification, adds up to be relevant for the annual mean mixed-layer DIC budget. These results support the hypothesis that storm-driven ocean physics is a significant participant in the Southern Ocean carbon cycle.

## Results

**High-resolution observations of mixing and $CO_2$ variability.** The subpolar Southern Ocean is where the upwelling of Upper Circumpolar Deep Waters (UCDW) supplies elevated concentrations of DIC to the base of the mixed layer[29] (Fig. 1). This gives rise to a circumpolar zonal band of spatially and seasonally varying outgassing of natural $CO_2$ (Fig. 1a, b). A surface autonomous vehicle (Wave Glider measuring atmospheric weather and carbon chemistry) and an ocean profiling vehicle (Slocum glider measuring physical characteristics of the water column) sampled one site (54°S, 0°E) in a coordinated way to yield an integrated high-resolution view of the interactions between the air-sea fluxes and upper-ocean dynamics in the South Atlantic Ocean (Fig. 1, refer to Methods).

The air-sea flux of $CO_2$ is defined (Eq. 1) as the product of the air-sea gradient of $pCO_2$ ($\Delta pCO_2 = pCO_{2sea} - pCO_{2atm}$, where $pCO_{2sea}$ is the partial pressure of $CO_2$ in the surface ocean and $pCO_{2atm}$ in the atmosphere), the temperature-dependent solubility $K_s$ and the wind dependent air-sea gas transfer velocity $K_w$[30].

$$F_{CO2} = K_w \times K_s \times \triangle pCO_2 \tag{1}$$

Notably, $K_w$ which is quadratic in wind speed[30] and $\Delta pCO_2$ are both crucial to $F_{CO2}$, but this study focuses primarily on how storms influence $\Delta pCO_2$. The direction and part of the magnitude of $F_{CO2}$ are set by $\Delta pCO_2$, the variability of which is dominated by $pCO_{2sea}$ in our observations (Supplementary Fig. 1). The observed $pCO_{2sea}$ and hence $\Delta pCO_2$ varied by ~±10 μatm, as $F_{CO2}$ oscillated between uptake and outgassing on synoptic timescales (1–10 days) (Fig. 1e). Several of the outgassing events coincided with the passage of storms (Fig. 1e—compare grey and red shaded areas). To put these results into perspective, the synoptic variability of $\Delta pCO_2$ (about 20 μatm from peak to trough) is similar in magnitude to the seasonal amplitude of $\Delta pCO_2$ and $pCO_{2sea}$ for the subpolar Southern Ocean[10,31].

To elucidate the causes of synoptic variability of observed $\Delta pCO_2$ and $pCO_{2sea}$, we begin by decomposing the drivers of changes in $pCO_{2sea}$ into the relative contributions made by thermal ($pCO_{2-SST}$) and by non-thermal ($pCO_{2-DIC}$) components[32] referenced from the start of the deployment (Fig. 1f). This decomposition shows two timescales of variability, which are key to explaining the mechanisms: first, the synoptic-scale variability of $\Delta pCO_2$ and $pCO_{2sea}$ (Fig. 1e, f) is dominated by the $pCO_{2-DIC}$ component, which we will show is primarily caused by wind-driven DIC transport in the ocean. In fact, the synoptic $pCO_{2-DIC}$ ($pCO_{2-DIC}'$, computed by removing the 10-day rolling mean from $pCO_{2-DIC}$) explains about 70% of the variations in $pCO_{2sea}$ ($r^2 = 0.71$, and refer to Figs. 1e, 2d, and Supplementary Fig. 1). Second, both $pCO_{2-DIC}$ and $pCO_{2-SST}$ also show summer seasonal trends, which are weakening for $pCO_{2-DIC}$ and strengthening for $pCO_{2-SST}$ (Fig. 1f). Over the duration of the deployment, these trends in $pCO_{2-DIC}$ and $pCO_{2-SST}$ ultimately yield changes of

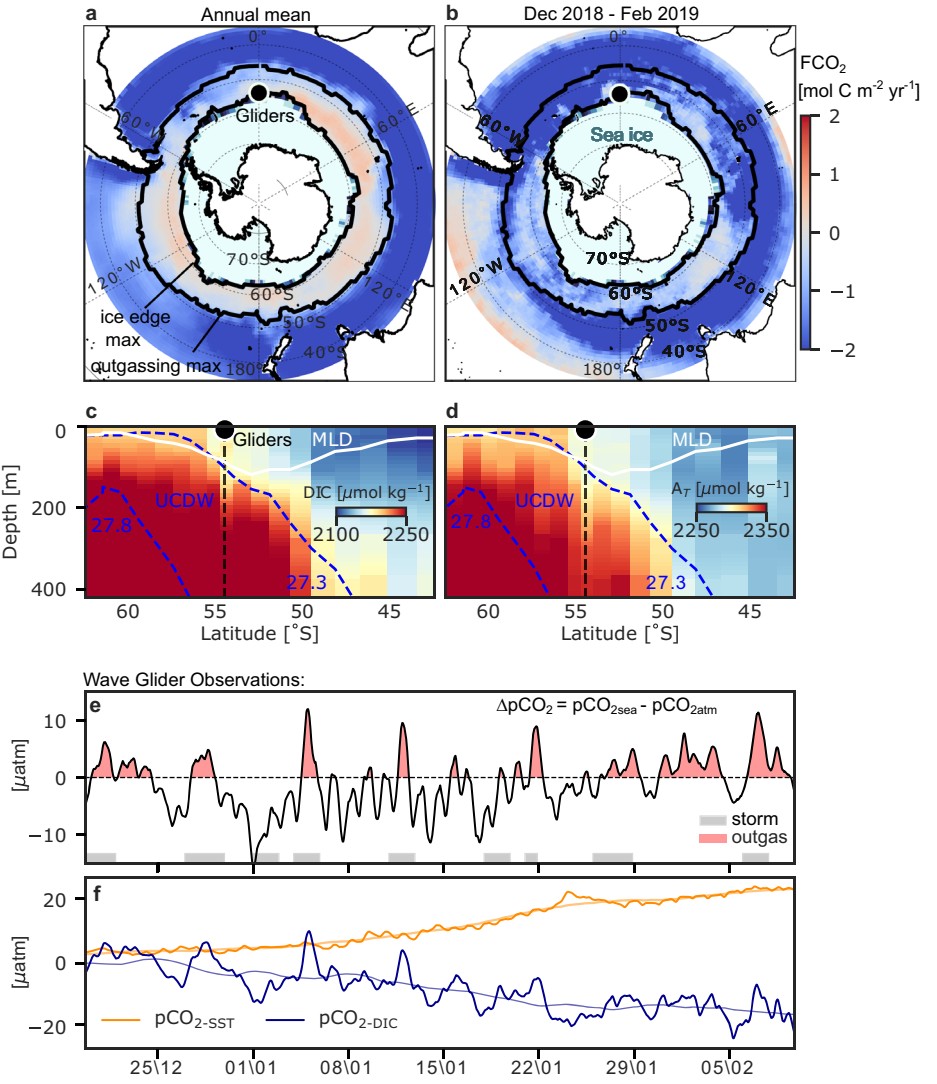

**Fig. 1 Observed temporal variability of ΔpCO₂ in the outgassing domain of the Southern Ocean. a** The annual mean (2005–2019) net air-sea CO₂ flux (F$_{CO2}$) [mol C m⁻² yr⁻¹] from CSIR-ML6[1,41]. Overlaid is the climatological sea-ice concentration maximum from NCEP-DOE AMIP-II Reanalysis 2[81]. The black dot marks the location of the robotic platforms (labelled Gliders comprising a Wave Glider and Slocum glider). The subpolar outgassing region considered here is between the climatological sea-ice-edge maximum and the extent of the zonal band of maximum outgassing for 2005–2019 determined by the 0 contour of the CO₂ flux in winter (June–August), shown by black contours. **b** is as in **a** except F$_{CO2}$ is averaged over only the Gliders deployment period (Dec–Feb 2019). An example meridional section of **c** Dissolved Inorganic Carbon (DIC) [μmol kg⁻¹] and **d** Total Alkalinity (A$_T$) [μmol kg⁻¹] along the Good Hope Line (transect AX25) during 2016 from GLODAPv2.2020[82,83]. The dashed blue contours (27.3 and 27.8 kg m⁻³ isopycnals) are the upper and lower bounds of the Upper Circumpolar Deep Water (UCDW). The white contour is the mixed-layer depth (MLD). **e** Wave Glider observed ΔpCO₂, which is the difference between the partial pressure of CO₂ in the surface ocean (pCO$_{2sea}$) and in the atmosphere (pCO$_{2atm}$) in μatm. Grey bars highlight the central part of a storm passage defined using the 25$^{th}$ sea level pressure and 75$^{th}$ wind speed percentiles (Supplementary Fig. 2). **f** Decomposition of pCO$_{2sea}$ into its thermal (pCO$_{2-SST}$) and non-thermal (pCO$_{2-DIC}$) drivers. The thick lines represent the cumulative contribution of each process to the observed changes in ΔpCO₂ relative to the start of deployment (time = 0). The thin lines show the 10-day rolling mean. Time is given as dd\mm of 2018 and 2019.

~20 μatm, which are comparable in magnitude to the synoptic variability (Fig. 1e). The increasing trend in pCO$_{2-SST}$ is due to progressively increasing sea surface temperature (SST) linked to seasonally driven solar warming (Supplementary Fig. 3a). Meanwhile, the weakening trend in pCO$_{2-DIC}$ is likely a consequence of a gradual decrease in DIC by ~10 mmol C m⁻³ (or equivalently 1000 mmol C m⁻² assuming the top 100 m is mixed) over 2 months due to biological productivity. Consistent with this inferred DIC drawdown of 1000 mmol C m⁻², bio-optical estimates of net primary productivity from sensors on the Slocum glider and satellites range from ~13–40 mmol C m⁻² d⁻¹ or equivalently 700–2300 mmol C m⁻² over the 56-day deployment

(Supplementary Fig. 4). Ocean advection and a net freshwater flux due to precipitation could also influence this trend in pCO$_{2-DIC}$ significantly (Supplementary Fig. 3b), but the relative contributions of advection and freshwater fluxes are not investigated in this study (refer to[31]). Regardless of the driving mechanisms, the seasonal trends in pCO$_{2-DIC}$ and pCO$_{2-SST}$ approximately compensate for each other, thus ΔpCO₂ fluctuates about its initial value near zero, and the sign of ΔpCO₂ changes on synoptic timescales during the 2-month glider deployment at this site.

To link the synoptic variability of ΔpCO₂ and pCO$_{2sea}$ (Fig. 1e, f) with wind-driven mixed-layer processes that connect the surface ocean to the DIC-rich subsurface reservoir (Fig. 1c), we show the

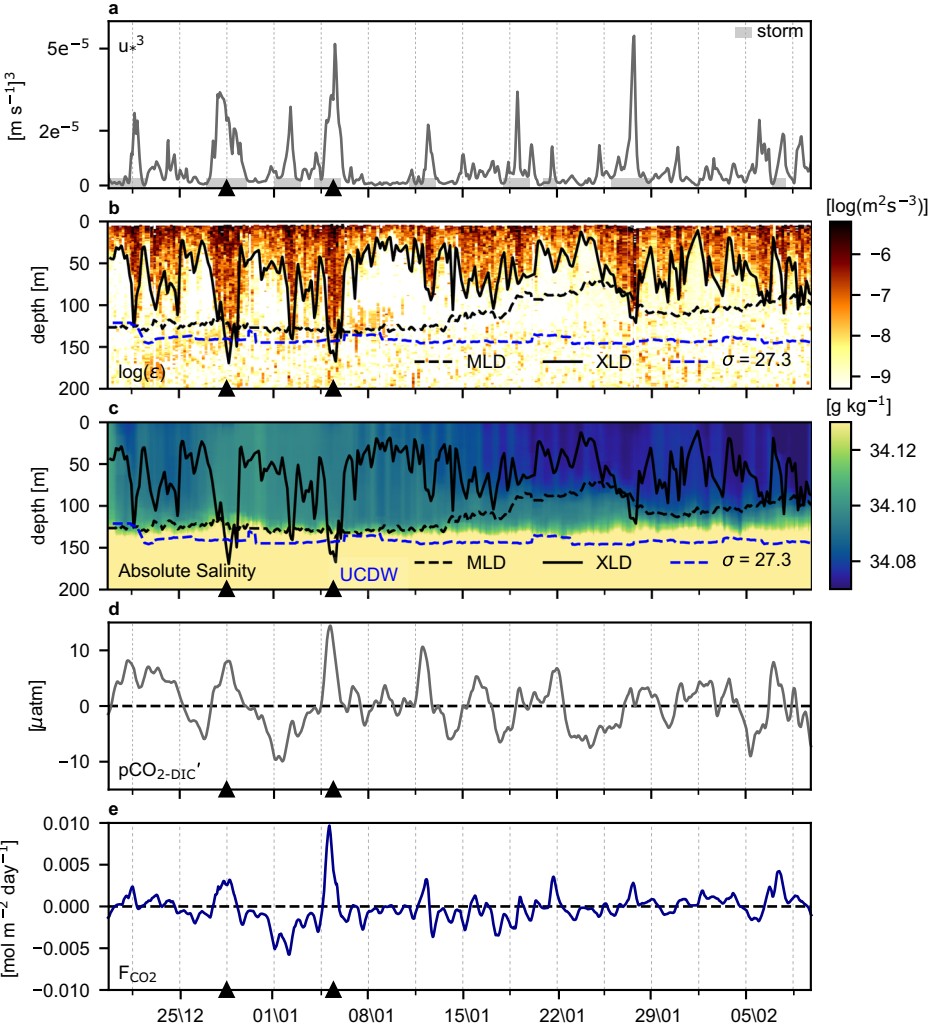

**Fig. 2 Linking wind to vertical mixing and surface ocean pCO$_2$ variability. a** Friction velocity (u$_*$) cubed [m s $^{-1}$]$^3$ computed from the Wave Glider (WG) wind speeds during austral summer 2018–2019. Grey bars highlight the presence of storms as defined Fig. 1e. Two prominent storm events are highlighted by black triangles. **b** Depth-time section of the upper-ocean dissipation rate of turbulent kinetic energy ($\varepsilon$)[m$^2$ s$^{-3}$] estimated from shear probes on an RSI MicroRider onboard the profiling glider. As shown in Supplementary Fig. 5, u$^{*3} \approx \varepsilon z$ where z is depth. The solid black line is the mixing-layer depth (XLD), the dashed black line is the mixed-layer depth (MLD) and the blue dashed line is the 27.3 kg m$^{-3}$ isopycnal indicating the upper bound of Upper Circumpolar Deep Water (UCDW). **c** Depth-time section of Absolute Salinity [g kg$^{-1}$] from the profiling glider with the XLD, MLD and UCDW isopycnal overlaid. **d** WG observed pCO$_{2-DIC}$' [µatm], which is the synoptic anomaly (computed by removing the 10-day rolling mean) of the non-thermal component of pCO$_2$. **e** WG observed CO$_2$ flux (F$_{CO2}$) [mol C m$^{-2}$ day$^{-1}$].

surface friction velocity due to the wind (u$_*$ defined in Methods) and the dissipation rate of turbulent kinetic energy ($\varepsilon$) observed by the profiling glider (Fig. 2a, b). The dissipation rate is a measure of turbulence and is related to turbulent diffusion and vertical mixing through stratification. The vertical extent of the elevated $\varepsilon$ is representative of the turbulent boundary layer depth, hereafter referred to as the mixing-layer depth (XLD, defined in ref. [33]). Turbulent, deep-reaching mixing events down to 150 m correspond with strong wind (high u$_*$) compared with periods of shallower (about 50 m), weaker mixing events occurring during weak winds (low u$_*$). Here, the observed variability in the magnitude ($\varepsilon$) and depth of mixing (XLD) was driven primarily by wind (u$_*$), r$^2$ = 0.76 and r$^2$ = 0.75, respectively (Supplementary Figs. 5 and 6). In contrast to the highly variable XLD, the density-derived mixed-layer depth (MLD, defined in Methods) was nearly constant through mid-January at around 130 m (Fig. 2b). This initial MLD of 130 m is set by the winter MLD maximum, and it is coupled to the upper bound of the high-salinity UCDW ($\sigma$= 27.3 kg m$^{-3}$, defined in ref. [34]). During mid-January the MLD shoaled to about 75 m, decoupling

from UCDW and its carbon-rich waters (Fig. 2b, c). The MLD was not directly sensitive to the variability of the wind (r$^2$ = 0, Fig. 2a, b) and was thus distinctly different from the XLD. This is consistent with our conventional density threshold definition of the MLD (refer to Methods), which is not expected to vary with the XLD on synoptic or shorter timescales[33]. The absence of correlation between the MLD and the wind is also consistent with the model of Whitt et al. (2019)[22], in which sub-seasonal u$_*$ is weakly correlated with sub-seasonal MLD both in the subpolar Atlantic sector of the Southern Ocean and globally. In particular, since the MLD is set by the integrated effect of mixing, it does not vary on the same timescales as the wind. Thus, the MLD and XLD manifest different mechanisms that dominate their modes of variability (seasonal vs synoptic).

But, how do the synoptic variations in pCO$_{2sea}$ relate to wind-driven upper-ocean physical variability? We address this question first via qualitative analysis of prominent wind events in the time series. During intense wind events (>20 m s$^{-1}$, Supplementary Fig. 2), when the XLD > MLD (particularly during two storm events centred on 28 December 2018 and 5 January 2019, Fig. 2), the

mixing vertically entrained the underlying UCDW with its high salinity and DIC into the mixed layer (since both salinity and DIC increase rapidly with depth at the top of the UCDW; Fig. 1c and Fig. 2c). As expected, the entrainment events are followed by increased salinity in the mixed layer (Fig. 2c), elevated $pCO_{2\text{-DIC}}'$ (Fig. 2d), and a reversal from ingassing to outgassing (Fig. 2e) that is indicative of higher DIC in the mixed layer. These two events are associated with two of the largest $CO_2$ outgassing peaks observed during the experiment (Fig. 2e) and shifted the mean sea-to-air $F_{CO_2}$ by 50% from $-0.12$ mol C m$^{-2}$ yr$^{-1}$ (i.e. the mean excluding these two events of positive $F_{CO_2}$) to $-0.06$ mol C m$^{-2}$ yr$^{-1}$ (i.e. the mean of the full record in Fig. 2e including these two events).

The two mixing events described above suggest that storm-driven mixing and entrainment provide a significant driver of transient outgassing. However, consideration of a later storm event reveals a more nuanced perspective. During the strong wind event on the 28 of January 2019, when the XLD > MLD (Fig. 2b) and the friction velocity reached its highest value during the multi-glider deployment (Fig. 2a), there was only a small increase in mixed-layer salinity (Fig. 2c) and $pCO_{2\text{-DIC}}'$ (Fig. 2d). However, the MLD had shoaled and was decoupled from the more saline and DIC-rich UCDW. This suggests that sub-seasonal entrainment of UCDW is not only dependent on the XLD exceeding the MLD but rather on the XLD exceeding the winter MLD maximum (MLD$_{max}$, located at the 27.3 kg m$^{-3}$ isopycnal, Fig. 2b), which ultimately sets the depth of the subsurface reservoir of DIC and salinity until the following winter. Thus, the associated wind forcing by passing storms after winter and into late summer requires sufficient intensity to reach this subsurface reservoir of DIC and result in $CO_2$ outgassing events. It is therefore plausible that progressively stronger buoyancy forcing in summer (e.g. Fig. 6c in Whitt et al. (2019)[22] and Supplementary Fig. 3) may weaken the intermittent vertical transport of DIC associated with strong synoptic mixing events and contribute (together with the requisite seasonal DIC decrease) to the decreasing trend of $pCO_{2\text{-DIC}}$ in Fig. 1f. Finally, we observed several other synoptic-scale reversals in $\Delta pCO_2$ between uptake and outgassing events that were not explained by enhanced wind, mixing and entrainment. Although the $\Delta pCO_2$ gradient was reversed during such events, the magnitude of the $CO_2$ outgassing flux was considerably less than the two entrainment events explained above due to the low wind speed and small $K_w$ in Eq. (1) (Fig. 2e). The question that therefore arises is what could explain these further synoptic variations in $pCO_{2\text{-DIC}}$ that do not obviously coincide with strong wind and entrainment? The following section provides insight to this question by invoking a dynamical model for wind-driven synoptic variability of $pCO_{2\text{-DIC}}'$ that is driven by both vertical entrainment and meridional Ekman transport.

**A dynamical model for synoptic variability of surface ocean $pCO_2$.** We establish a model that combines two upper-ocean responses associated with the passage of a storm: storm-driven vertical entrainment, which is defined here to occur when the XLD exceeds the MLD$_{max}$, and storm-driven meridional advection by the Ekman flow (Ekman advection), which drives the meridional displacement of waters in the wind-driven mixing layer. Before proceeding to the physical model, it is important to note that $pCO_{2sea}$ is not a conservative tracer and it can be estimated from DIC and total alkalinity (A$_T$) using an empirical function, G[35]. The conceptual framework is to describe synoptic variability of $pCO_{2\text{-DIC}}'$ in terms of synoptic variability of conservative DIC and A$_T$:

$$pCO_{2-DIC}' = G\left(Y_{Ek}(DIC, A_T) + Z_{ent}(DIC, A_T)\right) \quad (2)$$

$Y_{Ek}$ (DIC, A$_T$) represents the anomalies of DIC and A$_T$ due to north-south displacements of water masses from the time-integrated Ekman advection:

$$Y_{Ek} = \int -v_{Ek}dt \times \frac{\partial(DIC, A_T)}{\partial y} \quad (3)$$

Where the meridional Ekman velocity is defined by $v_{Ek} = -\frac{\tau_x}{\rho_{sw}\text{XLD}f}$, $\tau_x$ is the zonal surface wind stress, $\rho_{sw}$ is the reference density of seawater, and $f$ is the Coriolis parameter (or inertial frequency which is negative in the Southern Hemisphere by convention), XLD is the mixing-layer depth, which is assumed equal to the Ekman depth. $\frac{\partial(DIC, A_T)}{\partial y}$ is approximated by the meridional gradient of DIC and A$_T$ computed using climatological concentrations of A$_T$ and DIC derived empirically[36,37] (dashed line, Fig. 3a, b). The temporal variability of the gradients $\frac{\partial(DIC, A_T)}{\partial y}$ is neglected for simplicity, based on the modest spread in the observed gradients at a few different times (Fig. 3b) and the strong performance of the model (Fig. 3d). The synoptic anomalies of $Y_{Ek}$ were separated by removing the 10-day rolling mean (Supplementary Fig. 7).

$Z_{ENT}$(DIC, A$_T$) represents the DIC and A$_T$ anomalies due to time-integrated vertical entrainment, which is irreversible and non-negative (unlike Ekman advection). That is,

$$Z_{ent} = \int H \times \left(\frac{1}{\text{MLD}_{max}} \times \left(C_{deep} - C_{surf}\right) \times \frac{\partial \text{XLD}}{\partial t}\right)dt \quad (4)$$

where using $x = \frac{\text{XLD}}{\text{MLD}_{max}}$,

$$H = 1, \; if \, x > 1 \; and \; \frac{\partial \text{XLD}}{\partial t} > 0 \, ,$$

$$H = 0, \; if \, x < 1 \; or \; \frac{\partial \text{XLD}}{\partial t} < 0 \, ,$$

and MLD$_{max}$ and XLD are positive by convention. It may be noted that this model omits biological sources and sinks as well as many transport processes, including diffusion through MLD, lateral mixing and several lateral and vertical advective processes, all of which turn out to be less significant than Ekman advection in driving the observed synoptic variability of $pCO_{2\text{-DIC}}$.

In Eq. (4), MLD$_{max}$ is the maximum MLD, C$_{surf}$ and C$_{deep}$ are averaged concentrations of DIC or A$_T$ within the surface to MLD$_{max}$ and in the subsurface between the MLD$_{max}$ to ~20 m below, respectively (see ref. [38] for a derivation of the entrainment tendency term in a surface-layer average heat budget). The time series of DIC and A$_T$ anomalies associated with entrainment are calculated numerically and added to the Ekman anomalies to obtain the surface ocean $pCO_2$ anomaly (see Methods for more details on the model).

To anticipate the model behaviour, consider the input horizontal and vertical DIC and A$_T$ gradients in Eqs. 3–4 (Fig. 3a, b; Fig. 1c, d), in addition to the wind forcing and mixing reported above (Fig. 2, Supplementary Figs. 5 and 6). In particular, the subpolar Southern Ocean has vertical and meridional gradients in DIC and A$_T$, with higher DIC and A$_T$ at depth and further south (Fig. 1c, d). The increasing surface concentrations of DIC and A$_T$ to the south (Fig. 3b) are primarily a result of the large-scale meridional upwelling of UCDW (high in DIC and A$_T$) in the south (Fig. 1c, d) and the effect of southward decreasing sea surface temperature on $CO_2$ solubility[39]. At 54°S, 0°E, the DIC lateral and vertical gradients were stronger than A$_T$ (Fig. 3a, b). Thus, the storm-driven supply of DIC into the mixed layer is more sensitive than A$_T$, which is important given that DIC and A$_T$ impact $pCO_{2sea}$ in opposing ways. This is illustrated by Fig. 3c, a DIC-A$_T$ vector plot[40], which compares the impact of Ekman and entrainment transports on the relative change of DIC and A$_T$. The ratio of A$_T$/DIC sets the

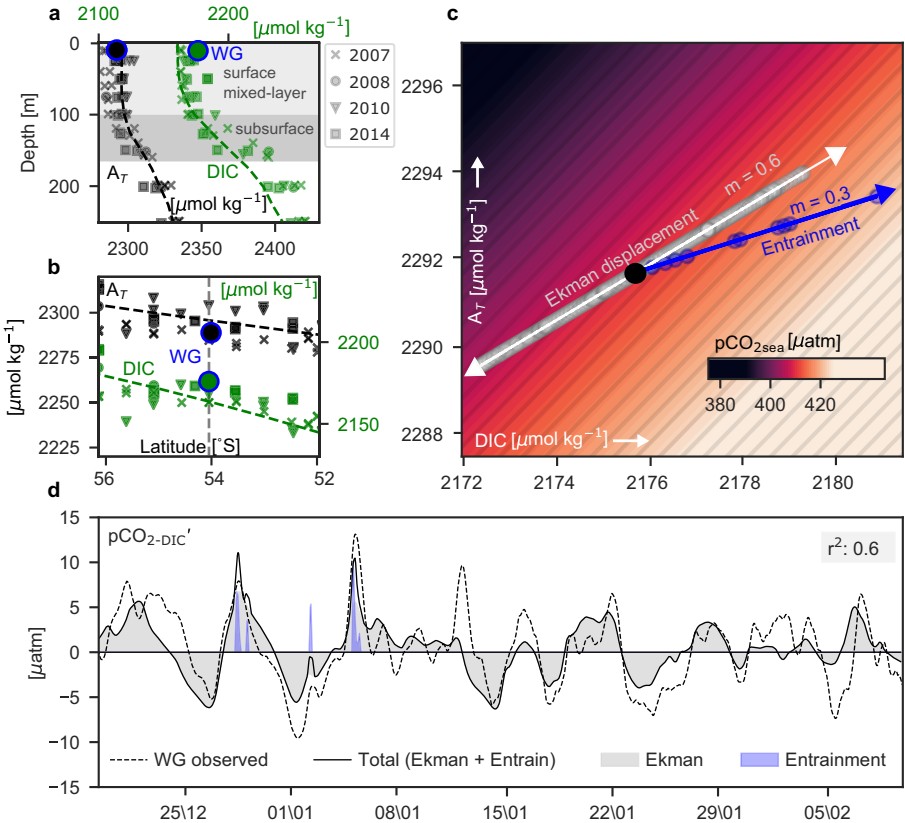

**Fig. 3 Conceptual model to explain observed synoptic variability in pCO$_{2\text{-DIC}}$'. a** Vertical profiles of total alkalinity (A$_T$, black) and dissolved inorganic carbon (DIC, green) for different cruises at the location of the Wave Glider (WG) (54 °S, 0 °E) taken from GLODAPv2.2020[82, 83]. **b** Lateral distribution of surface A$_T$ and DIC from the same cruises. Shown on both **a** and **b** are the average A$_T$ and DIC from the WG observations (blue outlined circles) (see Methods) and from the summer (Dec–Feb) gridded climatology (dashed lines)[36, 37]. **c** The range of WG estimated observations of surface A$_T$ (y-axis) vs DIC (x-axis) with the partial pressure of CO$_2$ in the surface ocean (pCO$_{2\text{sea}}$) as contours (µatm). Overlaid are the scatter plots of modelled lateral Ekman displacement of DIC vs A$_T$, the modelled entrainment of DIC vs A$_T$ and the corresponding slopes (m) are shown, and the time-averaged A$_T$ and DIC value estimated from the WG (black circle). **d** The black line shows the model estimate of the synoptic variability of the non-thermal component of pCO$_2$ (pCO$_{2\text{-DIC}}$') through combined Ekman advection and entrainment. The dashed line is the actual WG observed pCO$_{2\text{-DIC}}$'. A rolling mean of the local inertial period is applied to both and their corresponding coefficient of determination (r$^2$) is shown. The grey shading shows the separate estimated contribution due to Ekman advection and the blue shading of entrainment to estimated pCO$_{2\text{-DIC}}$'.

sensitivity of the pCO$_{2\text{sea}}$ to both Ekman and entrainment processes (Fig. 3c). If the ratio of A$_T$/DIC were about 1:1, the slope would be along the pCO$_{2\text{sea}}$ isolines and the impact of Ekman advection and vertical entrainment on pCO$_{2\text{sea}}$ would be negligible (Fig. 3c). Our model shows that the A$_T$ versus DIC slope (m) for entrainment is 0.3 and for Ekman transport is 0.6, revealing that the pCO$_{2\text{sea}}$ anomalies are positively correlated with DIC anomalies for both entrainment and Ekman processes, but more sensitive to DIC anomalies due to entrainment than to Ekman (Fig. 3c and Supplementary Fig. 8). Thus, the horizontal and vertical gradients of DIC and A$_T$ are important factors in determining how efficient these physical processes are in driving the synoptic variability of pCO$_{2\text{sea}}$.

The resulting time series of the estimated pCO$_{2\text{-DIC}}$' computed using the model (Eq. 2) is compared to the Wave Glider observed pCO$_{2\text{-DIC}}$' (Fig. 3d). Strikingly, the modelled pCO$_{2\text{-DIC}}$' reproduces most of the observed synoptic variability in pCO$_{2\text{-DIC}}$', accounting for 60% of the total observed variance in pCO$_{2\text{-DIC}}$'. However, there are some discrepancies between the estimated and the observed values, such as the phasing of the estimated variability is not always aligned with the observed pCO$_{2\text{-DIC}}$' (e.g. events on 25/12 and 05/02 had slightly delayed observed responses). Likewise, the estimated magnitude of the response is sometimes underestimated (12/01) or overestimated (30/12).

Nevertheless, we conclude that physical transport associated with Ekman advection and entrainment and encapsulated in the model is the dominant cause of the observed synoptic variability in pCO$_{2\text{-DIC}}$, ΔpCO$_2$ and pCO$_{2\text{sea}}$. All other transport and biological processes likely explain less than 40% of the variance and are therefore subdominant. For example, variability in biological sources and sinks of DIC may explain a small fraction of the synoptic variance in pCO$_{2\text{-DIC}}$, but the amplitude of synoptic variations in net primary productivity derived from in situ optical measurements are estimated to be an order of magnitude too weak to explain the observed synoptic variability in pCO$_{2\text{-DIC}}$ (Supplementary Fig. 9).

The model also allows for the separation of Ekman advection (grey shading) and entrainment (blue shading) contributions to pCO$_{2\text{-DIC}}$' variability (Fig. 3d). We find that storm-driven Ekman displacement dominates entrainment and explains most of the synoptic pCO$_{2\text{-DIC}}$, ΔpCO$_2$ and pCO$_{2\text{sea}}$ variations during the deployment, including during the strong entrainment events discussed previously (Fig. 3d). But entrainment does cause rare large pCO$_{2\text{-DIC}}$' anomalies and it contributes substantially to the strongest outgassing fluxes F$_{CO_2}$ observed on 28 December 2018 and 5 January 2019 that result from a synergistic combination of positive pCO$_{2\text{-DIC}}$' and ΔpCO$_2$ due to Ekman advection and entrainment as well as large K$_w$ from strong winds (Fig. 2d).

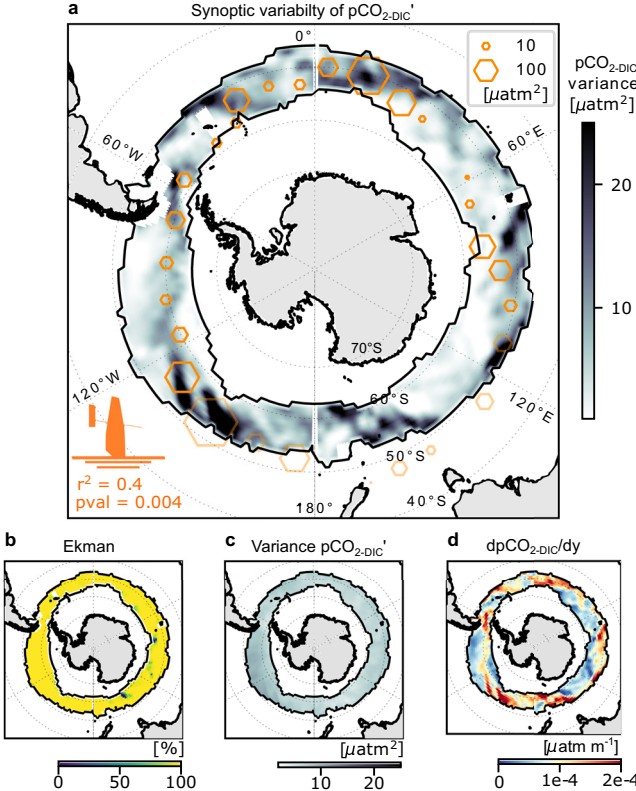

**Fig. 4 The spatial distribution of synoptic variance of pCO$_{2\text{-DIC}}$ in the subpolar Southern Ocean.** **a** Modelled 7-day variance of the synoptic anomalies in the non-thermal component of pCO$_2$ (pCO$_{2\text{-DIC}}$') [µatm]$^2$ computed and averaged for 2019 (color bar). Overlaid are hexagons of co-located 7-day pCO$_{2\text{-DIC}}$' variance [µatm]$^2$ as observed from a Saildrone that circumnavigated Antarctica in 2019[28, 42]. The spatial correlation r$^2$ and associated *p*-value of estimated versus observed are indicated. Hexagons outside of the subpolar domain are displayed with transparency and are excluded from the statistics. **b** the relative contribution of Ekman [%] to the synoptic pCO$_{2\text{-DIC}}$' variability shown in **a**. **c** shows the modelled 7-day pCO$_{2\text{-DIC}}$' variance [µatm]$^2$ as in **a**, instead computed using a spatially uniform gradient of total alkalinity (A$_T$) and dissolved inorganic carbon (DIC) in Eq. 3. Thus, comparing **a** with **c** shows that the spatial variability of pCO$_{2\text{-DIC}}$' is not driven by spatial variability in wind, but rather the spatial variability driven by spatially diverse meridional gradients of A$_T$ and DIC and thus non-thermal component of pCO$_2$ (pCO$_{2\text{-DIC}}$). This is further evidenced when comparing (a) with (d) the meridional gradients of pCO$_{2\text{-DIC}}$ [µatm m$^{-1}$]. Black contours on all panels show the location of the climatological sea-ice-edge maximum and the outgassing maximum for 2005–2019, as in Fig. 1.

## Discussion

In this study, we present observations of the coupled physical-carbon processes by which storms drive synoptic variability of ΔpCO$_2$ and brief, but strong, outgassing events in the subpolar SE Atlantic. Hence, the results from this process study raise important broader questions: how prevalent is this strong synoptic variability in the coupled ocean physics–carbon system around the entire subpolar Southern Ocean? And what does it mean for the larger spatial and seasonal-inter-annual CO$_2$ flux dynamics?

**Prevalence of storm-driven synoptic variability in pCO$_2$ across the subpolar Southern Ocean.** In order to shed light on the circumpolar prevalence of regions of synoptic variability and the larger-scale implications as well as to test the proposed Entrainment-Ekman conceptual model for pCO$_{2\text{-DIC}}$, we have applied the model (Eqs. 2–4) over the dynamically comparable circumpolar upwelling zone of the Southern Ocean (between the Polar Front and the northern limit of sea ice in winter; Fig. 4; see Methods). The magnitude of the estimated synoptic variance (i.e. mean square anomaly) of pCO$_{2\text{-DIC}}$ can reach 25 µatm$^2$ in places with a mean-variance of 6 µatm$^2$ (Fig. 4a). For perspective, this 6 µatm$^2$ variance due to storms is approximately half in the magnitude of the summer mean inter-annual variance of pCO$_{2\text{sea}}$ estimated for the subpolar region of 13 µatm$^2$ computed from the CSIR-ML6 observation-based product[1,41]. Moreover, ~30% of the surface area of the subpolar ocean has a synoptic variance that is greater or equal in magnitude to the summer mean inter-annual variance of pCO$_{2\text{sea}}$, thus synoptic variability of pCO$_{2\text{sea}}$ is potentially a widespread dominant mode of variability in the Southern Ocean.

To assess the robustness of the estimated spatial distribution of synoptic variability of pCO$_{2\text{-DIC}}$', we spatio-temporally collocate the estimate derived from Eqs. 2–4 with recent Saildrone Antarctic circumpolar pCO$_2$ observations[28,42]. We find a statistically significant agreement between our estimate of the pCO$_{2\text{-DIC}}$' variance and the observed spatio-temporal synoptic variability of pCO$_{2\text{-DIC}}$' (r$^2$ = 0.4, Fig. 4a) from the Saildrone dataset. Regions of higher variance in the modelled pCO$_{2\text{-DIC}}$', such as in part of the SE Atlantic, coincide with higher observed pCO$_{2\text{-DIC}}$' variance observed by the Saildrone (indicated by the size of the hexagon). In the western Pacific sector, low estimated synoptic-frequency variance overlapped with lower measured variance (Fig. 4).

In agreement with the in situ observations from the multi-glider deployment (see Fig. 3), lateral Ekman advection over vertical entrainment is the dominant physical driver of the estimated high-frequency variability explaining about 92% of the model estimated variance of pCO$_{2\text{-DIC}}$' across the subpolar Southern Ocean (Fig. 4b). Importantly, the spatial variation in the meridional gradients of pCO$_{2\text{-DIC}}$ explained most of the spatial variation in high-frequency temporal variability of pCO$_{2\text{-DIC}}$' (Fig. 4a, c, d). This is consistent with Ekman advection scaling with the zonal wind stress and meridional gradients of DIC and Total Alkalinity $\frac{\partial(DIC, A_T)}{\partial y}$, because the spatial variability of the high-frequency wind is relatively uniform across this region (synoptic atmospheric variability occurs across large spatial scales of order 1000 km, refer Fig. 4c).

The hypothesized dominance of Ekman advection of mean gradients as a driver of the synoptic variability of pCO$_{2\text{-DIC}}$ has some somewhat surprising and important implications. First, it implies that time-mean gradients $\frac{\partial(<DIC, A_T>)}{\partial y}$ are generally larger than anomalies $\frac{\partial(DIC, A_T)'}{\partial y}$ on the timescales relevant to the synoptic lateral advection (that is days to weeks). This result in turn implies that the large-scale (>500 km) A$_T$ and DIC fronts are fairly stable in time (e.g. as seen by the low meridional variability of the DIC and A$_T$ observed in Fig. 3b). Drivers of large-scale variability such as changes in the large-scale circulation, biological productivity and air-sea fluxes evidently do not cause substantial inter-annual or even seasonal deviations from the time-mean $\frac{\partial(<DIC, A_T>)}{\partial y}$. In addition, mesoscale (<500 km) DIC and A$_T$ gradients are also weak relative to the large-scale time-mean $\frac{\partial(<DIC, A_T>)}{\partial y}$. Stirring by mesoscale eddies is relatively ineffective at producing A$_T$ and DIC anomalies (e.g. via frontogenesis, see ref. 43) compared to the mechanisms that are destroying the mesoscale gradients. Relatedly, the dominance of Ekman advection also implies that Ekman velocities dominate all other sources of synoptic velocity variability in the mixed layer, that is

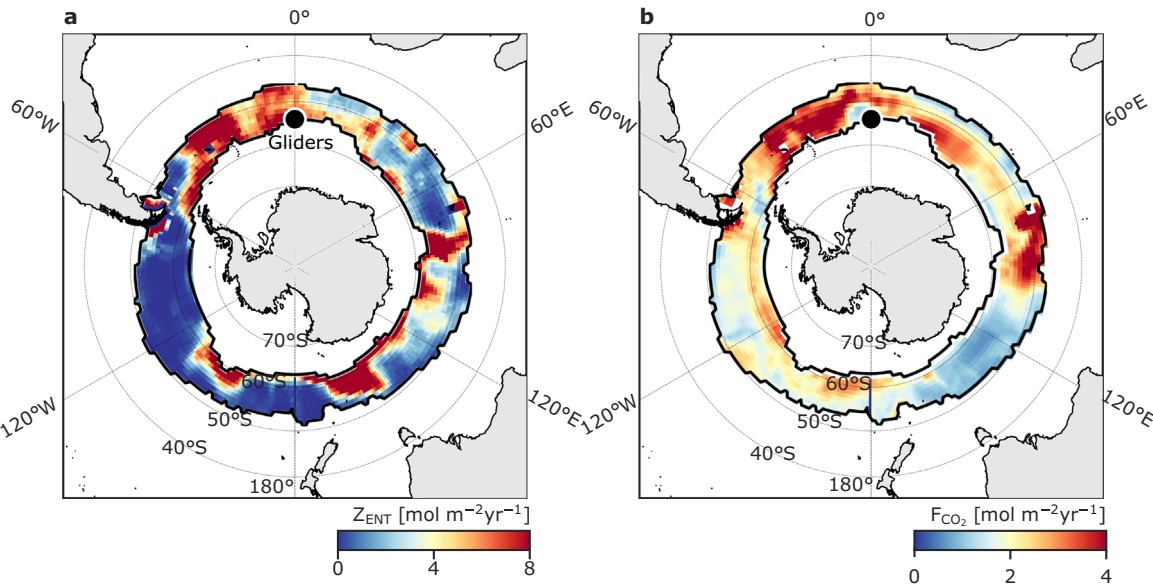

**Fig. 5 Storm-driven entrainment has the same order-of-magnitude effect on the mixed-layer DIC budget as the air-sea CO₂ flux across the subpolar Southern Ocean.** For perspective, we plot **a** the climatological annual mean entrainment flux ($Z_{ENT}$, see Eq. 4 and refer to Methods) of Dissolved Inorganic Carbon (DIC) [mol m⁻² yr⁻¹] in comparison with **b** the climatological seasonal amplitude of the seasonal cycle in the CO₂ flux ($F_{CO2}$) [mol m⁻² yr⁻¹] from CSIR-ML6[1, 41]. Both **a** and **b** are estimated over the period 2005–2019. Both **a** and **b** may be compared with the time-mean $F_{CO2}$ in Fig. 1a.

mesoscale and submesoscale turbulent velocities are relatively weak compared to Ekman velocities on synoptic timescales. A full explanation of these results and a broader evaluation of these hypotheses is beyond the scope of this work, but it is important to recognize that the explanatory power of Ekman advection of mean gradients and hence the accuracy of the extrapolations in Fig. 4 depend on the relative weakness of both variability in meridional gradients of pCO₂₋DIC and non-Ekman synoptic velocities, as inferred from observations in the SE Atlantic multi-glider deployment. We have not investigated what sets the magnitude of these large-scale mean lateral gradients of pCO₂₋DIC (Fig. 4d) or the mesoscale kinetic energy; however, these topics have been explored in other papers and are thought to be associated with the large-scale ocean fronts of the Antarctic Circumpolar Current (e.g. refs. [44–46]).

**Implications of storm-driven synoptic variability for carbon dynamics on longer timescales.** Another outstanding question is whether or not the storm-driven Ekman advection and entrainment have implications for the slower seasonal or inter-annual carbon dynamics of the Southern Ocean? Here, we address this question in two parts, first focusing on the Ekman advection and then entrainment. Perhaps the most striking result derived from the glider deployment and the subsequent extrapolation across the subpolar Southern Ocean is that a simple Ekman advection of mean meridional gradients explains the majority of the synoptic variance in pCO₂ₛₑₐ and ΔpCO₂ (Fig. 3 and Fig. 4). However, to a first approximation (and by definition in Eq. 3), oscillatory synoptic perturbations to Ekman advection are reversible and do not sum to impact the mixed-layer DIC budget and $F_{CO2}$ on longer timescales. To this level of approximation, perturbations to Ekman advection only locally modulate the sign of ΔpCO₂ where the seasonal drivers associated with thermal and non-thermal effects yield small mean |ΔpCO₂| relative to the synoptic perturbations, and thus the sign of ΔpCO₂ and the resulting $F_{CO2}$ is highly sensitive to the synoptic perturbations and varies on synoptic timescales. In addition, the observed mean ΔpCO₂ is only sensitive to the synoptic perturbations in Ekman advection

to the degree that the synoptic variance is also relatively large compared to the number of synoptic events or the duration of sampling (i.e. standard errors are large). However, oscillatory Ekman advection can be rectified in other subtle ways that are not captured at this level of approximation. For example, oscillatory advection of otherwise static ocean pCO₂ₛₑₐ spatial gradients under a spatially and temporally variable atmosphere may modify the average $F_{CO2}$ over longer time intervals due to correlation between Ekman-driven ΔpCO₂ anomalies and wind speed as well as the non-linear dependence of $F_{CO2}$ on wind speed. In addition, the combination of oscillatory Ekman advection and intermittent ocean mixing (e.g. during entrainment events) may induce lateral mixing via shear dispersion[47] that irreversibly sums to impact the slower evolution of the mixed-layer pCO₂ₛₑₐ. However, it is beyond the scope of this manuscript to quantify these rectified effects of Ekman advection for the large-scale dynamics of DIC and the air-sea CO₂ flux.

On the other hand, entrainment events, which have been shown to have a much smaller contribution to the synoptic variance than Ekman advection (Figs. 3 and 4a, b), connect the subsurface carbon-rich UCDW to the mixed layer irreversibly. Thus, all entrainment events sum to impact the mixed-layer DIC budget and hence $F_{CO2}$ on longer timescales (seasonal to inter-annual). The probability of sampling short storm-driven entrainment events like those observed by the multi-glider deployment with a 10-day (e.g. floats) or greater sampling period (e.g. ships) is very low and the response to entrainment is obscured by Ekman advection in any case. Thus, it is difficult to observationally quantify intermittent synoptic entrainment fluxes across the entire subpolar Southern Ocean as we do in the SE Atlantic with this data from paired gliders; coarser spatio-temporal sampling may alias this variability[17,28]. Hence, we use the model Eqs. 2–4 to provide an estimate of the magnitude of the time-averaged synoptic entrainment flux across the subpolar Southern Ocean. Figure 5 quantifies the annual mean entrainment flux of DIC (Eq. 4, see also Methods) and compares it (for perspective) with the magnitude of the climatological seasonal amplitude of $F_{CO2}$ (see also Fig. 1a, b). It shows that mean storm-

driven entrainment flux ($\sim3.5\,\mathrm{mol\,C\,m^{-2}\,y^{-1}}$) is of a similar order of magnitude to the amplitude of the seasonal cycle in $F_{CO2}$ ($\sim2.1\,\mathrm{mol\,C\,m^{-2}\,y^{-1}}$, peak to trough) as well as the time-mean $F_{CO2}$ (Fig. 1a; $\sim-0.1\,\mathrm{mol\,C\,m^{-2}\,y^{-1}}$). Hence, even small variations in the synoptic entrainment flux of DIC have a large impact on the mixed-layer DIC budget relative to $F_{CO2}$. If changes in synoptic entrainment go uncompensated by changes in other sources/sinks of mixed-layer DIC such as biological export production (which is plausibly of the same order of magnitude and opposite sign; see Supplementary Fig. 4) or other physical transport processes, the entrainment will drive changes in DIC and $F_{CO2}$. Consideration of the spatial structure of synoptic entrainment in Fig. 5 shows that entrainment exhibits substantial spatial variability and is particularly strong in the South Atlantic where the $\mathrm{MLD_{max}}$ is comparatively shallow relative to the Pacific basin (ref. [22] their Fig 10c) and storms are more frequent and stronger[13]. The spatial variability of the synoptic entrainment highlights the variable circumpolar implications of the observation reported above that storm-driven entrainment is sensitive to the winter MLD maximum (which sets the depth of UCDW reservoir) (Figs. 1b, c and 2b) and to stabilizing buoyancy forcing (which prevents storm-driven vertical mixing from reaching the UCDW reservoir during later summer months, e.g. in Fig. 2). Finally, these results emphasize that entrainment, which depends on the vertical gradients of DIC and $A_T$ and the MLD (Eq. 4), exhibits a quite different spatial structure than synoptic variance due to Ekman advection, which depends on the meridional gradient in $pCO_{2\text{-DIC}}$ (Eq. 3). Nevertheless, in both processes, different underlying oceanic conditions (in addition to the atmospheric conditions) are crucial determinants of the oceanic response to storms.

Although a complete analysis of the mixed-layer DIC budget is beyond the scope of this paper, the results in Fig. 5 indicate that it is possible that storm-driven entrainment cumulatively impacts the mean $pCO_{2\text{sea}}$ and $CO_2$ flux and influences inter-annual[28] and spatial variability of $CO_2$ outgassing (Fig. 1 and Fig. 5) through the interactions between annual changes in storm-characteristics[48], the seasonal cycle of the mixed layer, and variations to the depth of the winter MLD maximum[49]. We leave tests of these hypotheses to future work. The success of the conceptual model (Eqs. 2–4) in describing observations in the Atlantic sector (Fig. 3) coupled with the spatially variable and significant implications of storms for the mixed-layer DIC budget on a range of timescales (Figs. 4 and 5) motivate future field experiments and comprehensive modelling that can accurately quantify and predict the physical-carbon dynamics of the ocean mixed layer down to synoptic timescales more broadly around the subpolar Southern Ocean.

## Methods

**Experimental design and region of interest.** The basis for the observations conducted in this study is the SOSCEx-STORM experiment, which aims to simultaneously observe the passage of storms, their associated surface wind stress and corresponding upper-ocean response in physics and $CO_2$. The SOSCEx-STORM experiment forms part of a larger observational program, the Southern Ocean Seasonal Cycle Experiment (SOSCEx), detailed in ref. [50]. In SOSCEx-STORM, twinned gliders, a surface Wave Glider and a profiling glider, were programmed to sample together, allowing for a high-resolution view of the coupled atmosphere and upper-ocean processes. The field site chosen was at 54°S, 0°E, which is located south of the Antarctic Polar Front and in the globally significant outgassing sector of the Southern Ocean (Fig. 1). The platforms sampled for ~2 months (56 days), between 18 December 2018 and 12 February 2019.

### Autonomous observing platforms

*Wave Glider integrated with surface $CO_2$ sensor.* The Liquid Robotics SV3 Wave Glider (WG) was fitted with an Airmar WX-200 Ultrasonic Weather Station mounted on a mast at 0.7 m above sea level and sampled wind speed and direction at a rate of 1 Hz, averaged into 10 min bins. The surface winds were corrected to a height of 10 m above sea level as in ref. [51]. The WG was equipped with a SeaBird

Glider Payload CT-cell, measuring surface ocean temperature and conductivity at 1 Hz, averaged into 20 min bins. In addition, the WG was fitted with a VeGAS-pCO2 (Versatile Glider, Atmospheric and Ship pCO2 high Precision pCO2 analyzers) measuring atmospheric and ocean pCO2. The VeGAS-pCO2 sensor is based on the well-established NDIR (Licor - Li-820) linked equilibrator units[52,53] but with a significant redesign to improve accuracy (<1 µatm), precision (<1 µatm) through more effective drying and temperature control, equilibrator design and long term stability that also reduced the frequency of reference gas calibration from every sample to every 2 h. The unit was installed and linked to the SV3-WG control unit which enables remote communication and to send real-time data. These instruments have just recently been successfully assessed in the ICOS Ocean Thematic Centre instrument intercomparison study and those results will be published through ICOS. Outlier data points were removed from the WG pCO2 by applying a global cut-off upper 99.9 percentile and lower 0.1 percentile of the discrete temporal difference. A rolling mean with a window size of half the local inertial period (i.e. about 8 h) was applied to pCO2 WG data.

*Slocum integrated with an RSI MicroRider.* The water column observations were collected using a Teledyne Webb Slocum G2 glider. This profiling glider was equipped with a pumped SBE conductivity and temperature sensor, and a Rockland Scientific (RSI, Canada) MicroRider for microstructure measurements. The MicroRider was fitted with two piezo accelerometers and two air-foil shear probes and only collected data during the Slocum climbs. This choice was motivated by the need to increase the battery endurance by sampling for only half of the duration (about 3 months), and the choice of climbs over dives was made to ensure dissipation estimates as close to the surface as possible. The Slocum dataset was processed with the GEOMAR MATLAB toolbox and post-processing gridding (i.e. vertical interpolation to 1 m bins) and quality control of the glider data was carried out with GliderTools[54]. A Savitzky-Golay[55] filter was applied to the glider salinity and temperature profiles to remove spurious spiking in the data (as recommended in GliderTools) and smoothed further with a rolling mean (9 profiles or about 18 h window). The GEOMAR toolbox includes a hydrodynamic glider flight model that produces a time series of flow past the sensor and angle of attack (AOA) which are required for the processing of accurate microstructure measurements (ref. [56], section 4). The microscale velocity shear was obtained from the shear probes measuring the two orthogonal components of the shear along the axis of the instrument ($\frac{\partial v}{\partial x}$ and $\frac{\partial w}{\partial x}$). The MicroRider accelerometers, which obtain high accuracy measurements of mechanical-driven or impact-driven vibrations felt by the Slocum glider, were used to remove vehicle vibration contamination of the shear data[57]. The shear data from both probes were then processed into dissipation rate (ε) estimates using a four second FFT length and 12 second averaging[56]. The processing of the microstructure data was based on the routines provided by RSI (ODAS v4.04 software). The dissipation rate values were calculated assuming isotropic turbulence $\varepsilon = 7.5\nu\overline{\left(\frac{\partial w}{\partial x}\right)^2}$ (here written for the $w$ component) where $\nu$ is the seawater viscosity. The small-scale shear variance (the term with the overline) was obtained by integrating the wavenumber spectrum of shear in a wavenumber range that is relatively unaffected by noise and corrected for the unresolved variance using the empirical model from ref. [58]. The dissipation estimates underwent further quality controls. This included, globally, any segments with AOA larger than 6°, pitch larger than 30°, and Figure of Merit (a measure of spectra fit the Nasmyth spectrum, high values are a poorer fit) larger than 2 are excluded[56] (RSI Technical note 039, https://rocklandscientific.com/support/knowledge-base/technical-notes). We further excluded individual dissipation estimates which seemed unrealistic in value (e.g., data spikes due to plankton interactions on the shear probes).

### Derived metrics

*Friction velocity and theoretical dissipation.* Wind observations from the Wave Glider and dissipation estimated from shear probes on the MicroRider are used to investigate the link between wind and ocean turbulence via the friction velocity $u*$ defined as, $u* = \sqrt{\frac{\tau}{\rho_{sw}}}$, where $\rho_{sw}$ is the density of seawater and $\tau$ is the wind stress estimated using[59]. Similarity theory states that wind, through $u*$, impacts the dissipation rate of turbulent kinetic energy (ε) via $\varepsilon = \frac{u*^3}{kz}$, where $k = 0.41$ is the von Karman's constant and $z$ is the depth range below the surface ocean[60].

*Mixed-layer depth and Mixing-layer depth.* The mixed-layer depth, MLD, was calculated as the depth from the surface where the density first exceeds its surface value by $0.03\,\mathrm{kg\,m^{-3}}$ as in refs. [61,62], using the glider observed density sections. The mixing-layer depth, XLD, was estimated using the MicroRider derived ε profiles, as the depth from the surface where ε first drops below $10^{-8}\,\mathrm{m^2\,s^{-3}}$, following[33].

*Wave Glider estimated $A_T$, DIC and $CO_2$ flux.* The WG observed surface ocean temperature and salinity were used to estimate $A_T$[63], DIC was derived from estimated $A_T$ and WG observed $pCO_{2\text{sea}}$ using PyCO2SYS[64] (with options for equilibrium constants as follows: K1 and K2 are from Mehrbach, refit by Dickson and Millero, KSO4 from Dickson and TB from Uppstrom). Uncertainties in derived $A_T$ and DIC are about 5–10 µmol kg$^{-1}$. WG observed temperature, salinity and wind speeds along with the Japanese 55-year Reanalysis (JRA-55-do[65]) atmospheric sea level pressure (Supplementary Fig. 2) were used to compute air-sea $CO_2$ fluxes

using the bulk formulation with python package Seaflux.1.3.1 (https://github.com/lukegre/SeaFlux)[66].

*Net primary productivity*. Net primary productivity (NPP) is estimated from satellite ocean color (Ocean Colour-CCI[67,68]) using three different primary models: the Carbon-based Productivity Model (CbPM[69]), the Vertically Generalized Production Model (VGPM[70]) and the Platt model[71]. In addition, NPP is estimated from bio-optical measurements on the Slocum glider, which was fitted with a WETLabs ECO puck™ (BB2Fl-470/700), thus measuring chlorophyll-*a* fluorescence (proxy for phytoplankton concentration) and two wavelengths of optical backscattering by particles, bbp(470) and bbp(700). The optics data were cleaned and processed following procedures recommended by GliderTools[54]. Backscatter, bbp(700), was converted to phytoplankton carbon following Behrenfeld et al.[72]. Slocum glider NPP was estimated from the phytoplankton carbon, chlorophyll and collocated photosynthetically available radiation (PAR) from MODIS (the Slocum glider did not have a PAR sensor) following the CbPM[69] using python code PrimaryProductionTools.py (https://github.com/isgiddy/roammiz-seaice-impacts-organic-carbon/blob/v0/src/PrimaryProductionTools.py)[73] adapted from Arteaga et al.[74].

**Generalization of the conceptual model across the subpolar outgassing region**. We have applied the box model for high-frequency variability of surface ocean pCO$_2$ (defined in Eqs. 2–4) across a dynamically representative zonal band, which we define to be constrained to (1) a region that is not impacted by sea ice, (2) a region of mean outgassing, (3) a region with similar mean wind forcing and (4) mean mixed-layer depths. We used 3-hourly reanalysis winds to compute u$_*$ using winds from the Japanese 55-year Reanalysis (JRA-55-do[65]); monthly mean MLD were estimated using a density threshold of 0.03 kg m$^{-3}$ from density field derived from EN4.2.1 interpolated fields of temperature and salinity[75]; and temporal varying (i.e. monthly mean) lateral and vertical gradients from the climatology of A$_T$ and DIC were taken from refs. [36,37]. We translate wind variability via friction velocity (u$_*$) into mixing-layer depth (XLD) variability assuming the strong relationship between u$_*$ and XLD holds true for these dynamically comparable regions (Supplementary Figs. 5 and 6). Intermittent high-frequency supplies of DIC and A$_T$ due to high-frequency Ekman advection and wind-driven vertical entrainment were computed following (Eqs. 2–4) and the anomalies of DIC and A$_T$ were iteratively added to baseline climatological means of DIC and A$_T$. Changes in temperature were not included as the focus was on non-thermal drivers. Finally, PyCO2SYS[64] was used to compute pCO$_{2\text{-DIC}}$ from the physically-driven changes in A$_T$ and DIC to generate high-frequency temporal variability of pCO$_{2\text{-DIC}}$ anomalies across the Subpolar Southern Ocean (as shown in Fig. 4).

It is important to recognize that the generalization of the conceptual model (Eqs. 2–4) across the subpolar Southern Ocean for the full year (Fig. 4 and Fig. 5) is not validated by in situ observations at locations and times beyond those isolated validations reported here, and there are several assumptions that underpin the model. In particular, the model omits many potentially significant processes to isolate those that we find to be most important in the SE Atlantic. For example, non-thermal (pCO$_{2\text{-DIC}}$) components of pCO$_{2\text{sea}}$ are also driven by processes that include net community production (NCP), calcification, air-sea CO$_2$ exchange and freshwater fluxes, as well as various advective and mixing processes other than Ekman and entrainment, the influence of which is not included in the model. The premise for these omissions is that these terms have a comparably smaller impact on pCO$_{2\text{-DIC}}$ on these shorter 1–10-day timescales, as observed in the SE Atlantic during summer (Fig. 3d, Supplementary Fig. 9). Particular caution must be exercised when using the results of the conceptual model in regions where the neglected factors may be more significant, either because the strength of one or more of the neglected factors is greater or because the variability driven by Ekman and entrainment is weaker. With regard to the neglected NCP, for example, NPP is much stronger and more variable than at 54°S, 0°W in some localized regions surrounding subpolar islands and off the coast of South America (Supplementary Fig. 4). Conversely, physical DIC variability due to Ekman advection is substantially weaker in areas where meridional DIC gradients are weak (Fig. 4d). In addition, even the included entrainment process may not be well represented by the model at all times. For example, neglected effects of convection may have a stronger influence during winter than is reflected in the empirical relationships based on summertime observations (Supplementary Figs. 5 and 6). Furthermore, the magnitude and vertical extent of upper-ocean mixing and entrainment are influenced by other physical processes such as wave action[76,77] and submesoscale dynamics[78–80], which are not included. Finally, the generalization of our conceptual model is dependent on the accuracy of both reanalysis winds (JRA-55-do reanalysis has been shown to perform well in this region with a mean difference of −0.02 m s$^{-1}$ ± 0.8 m s$^{-1}$ with the WG wind speeds and refer to Supplementary Fig. 2 [65]) and on statistically inferred products of climatological DIC and A$_T$[36,37].

## Data availability

The data generated by this study have been placed in the Zenodo database and are available at https://doi.org/10.5281/zenodo.5674581. The Saildrone pCO$_2$ data used to support the findings of this study are available at https://doi.org/10.25921/6zja-cg56. The CSIR-ML6 CO$_2$ flux data are available from https://doi.org/10.25921/z682-mn47. The

total alkalinity and dissolved inorganic carbon observations used in this study are available as estimated monthly climatologies from https://doi.org/10.5194/essd-11-1109-2019 and https://doi.org/10.20350/digitalCSIC/10551, respectively, and as individual cruise transects from GLODAP version 2.2020 at https://doi.org/10.25921/2c8h-sa89. Surface chlorophyll-*a* are available from the OceanColour-CCI dataset (version 5) at https://doi.org/10.5285/1dbe7a109c0244aaad713e078fd3059a. EN4 quality controlled subsurface ocean temperature and salinity profiles (version 4.2.1) are available from https://www.metoffice.gov.uk/hadobs/en4/download-en4-2-1.html. Meteorological data from the Japanese 55-year Reanalysis (JRA-55-do) data are available at https://esgf-node.llnl.gov/search/input4mips/. To search, select "Target MIP" = "OMIP", "Institution ID" = "MRI", and "Source Version" = "1.4.0" among tabs on the left side. The sea-ice concentration data from NCEP_Reanalysis 2 data provided by the NOAA/OAR/ESRL PSL, Boulder, Colorado, USA, are available from their Website https://psl.noaa.gov/data/gridded/data.ncep.reanalysis2.gaussian.html.

## Code availability

The codes used for generating the main figures in the manuscript are available on GitHub at https://github.com/sarahnicholson/SouthernOceanStormsCO2 and are citeable https://doi.org/10.5281/zenodo.5675778.

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

## Acknowledgements

We are grateful to all scientists and support staff who helped in data collection and analysis of samples, in particular Sea Technology Services (STS) for technical assistance with glider deployments and SANAP, the captain and crew of the S.A. Agulhas II for their fieldwork/technical assistance. We thank Gerd Krahmann at Geomar for assistance with software tools for processing Slocum data, Isabelle Giddy and Thomas Ryan-Keogh for their assistance with estimating primary production from Slocum and satellite data, and Rockland Scientific Industrial for the technical support during the MicroRider deployment. We acknowledge the Centre for High-Performance Computing (CSIR-CHPC) for the support and computational hours required for the analysis of this work. This work is supported by CSIR Parliamentary Grant (SNA2011112600001); SANAP grants SNA170522231782, SNA170524232726 and SANAP200324510487; STINT-NRF Mobility Grant (STNT180910357293). S.S. is supported by a Wallenberg Academy Fellowship (WAF 2015.0186) and the Swedish Research Council (VR 2019-04400). M.d.P. is supported by the European Unions Horizon 2020 research and innovation programme under grant agreement No. 821001 (SO-CHIC). D.B.W. is supported by the National Science Foundation (NSF), via Grants OPP-1501993 and OCE-1658541, the National Oceanic and Atmospheric Administration, via Grant NA18OAR4310408, and the National Aeronautics and Space Administration, via NASA's Earth Science Research and Analysis Program (80NSSC19K1116). This material is based on work supported by the National Center for Atmospheric Research (NCAR), which is a major facility sponsored by the National Science Foundation under Cooperative Agreement No. 1852977. Travel funding from the Climate and Global Dynamics Laboratory at NCAR helped support this collaboration. The views, opinions, and findings contained in this report are those of the authors and should not be construed as an official NASA or U.S. Government position, policy, or decision. I.F. was supported by the Research Council of Norway via grant 294396. This is PMEL contribution 5231.

## Author contributions

S.N. performed the core analysis and figures (except those aspects described below) and wrote the manuscript. P.M.S. conceived the experimental idea and approach as well as contributed to writing. D.B.W. was instrumental in the theoretical interpretation and design of the dynamical model, as well as post-processing analysis of microstructure data and reanalysis data. I.F. led and assisted with the processing of microstructure dataset and contributed to its analysis and interpretation, M.d.P. conducted an analysis of seasonal tendency terms for salinity and temperature, A.D.L. assisted in processing Wave Glider $CO_2$ data, A.J.S. provided the Saildrone $pCO_2$ data. S.S. was involved in carrying out the experimental design and data collection. All authors contributed to revising the draft manuscript.

## Competing interests

The authors declare no competing interests.
