## [Peer Review File · Nature Communications]

REVIEWER COMMENTS

Reviewer #1 (Remarks to the Author):

The manuscript by Nicholson et al. is built on an impressive dataset that combines an array of novel observing technologies that have been applied in a very interesting study. The major focus of this study is a demonstration in the sub-of significant variability in surface ocean pCO₂ that is driven by storms. I.e., “air-sea gradient of CO₂ is dominated by synoptic storm-driven ocean variability” (lines 18/19). The data set is focused on two months of in situ pCO₂ measurements collected by a CO₂ sensor on a waveglider that operated during austral summer. I am not completely convinced of the major conclusions, however.

While this is quite a detailed analysis, I’m not sure that the basic premises are appropriate. The works was done near 54S, south of the Polar Front. This is a zone where the seasonal cycle of pCO₂ is driven by biological processes. The graphic pasted below shows a Hovmoller diagram of the annual cycle of pCO₂ in the South Atlantic, based on the climatology produced by the late Taro Takahashi (here made using ODV software and the data file downloaded from the ODV website). In the subtropics, the pCO₂ annual cycle is driven by temperature, with highest values in summer. South of the 40S, however, biological uptake of inorganic carbon dominates over the effects of temperature. The annual cycle inverts, relative to the subtropics, due to biological uptake of DIC during spring and summer that reduces pCO₂ more than the temperature effect on solubility increases pCO₂. This has been well recognized in numerous assessments. Yet, the analysis in the manuscript is focused on the balance of temperature effects driven by changes in CO₂ solubility (pCO₂-sst) and a pCO₂-DIC component that is “primarily caused by wind-driven DIC transport in the ocean”. The role of the major driving force, inorganic carbon uptake during photosynthesis, and the variability that this might drive and which dominates the seasonal cycle of pCO₂, is just not considered explicitly.

Given that biological uptake is the process that really drives the summer balance of CO₂ flux, I wouldn't think that (lines 319/321) “the sign of $\Delta p\text{CO}_2$ is dependent on synoptic (1-10 day) variability (approximately 20 μatm ; Fig. 1b) and thus key to constraining the mean FCO₂ over the summer season.” The variability that must be driven by biology is apparent in the data via their statements such as, “we observed several other synoptic-scale reversals in $\Delta p\text{CO}_2$ between uptake and outgassing events that were not explained by enhanced wind, mixing, and entrainment.” Biological uptake of carbon and reductions in pCO₂ are also likely responsible for lags between wind forcing and pCO₂ changes (“events on 25/12 and 05/02 had slightly delayed observed responses”).

In summary, I believe that this is a great set of observations. The two events (28 Dec, 5 Jan) nicely show the effects of deepened mixing on CO₂ flux. On the other hand, Figure 2e doesn't seem to show that those two events would dominate the integrated flux over the observational period. As a result I wouldn't necessarily be convinced by the manuscripts conclusions.

Other comments:

There needs to be more information on the pCO₂ sensor used in the paper. The one cited reference near the mention of the sensor on line 452/453 provides no information on the instrument.

Alkalinity estimates based were on Lee 2006. There are much more comprehensive tools, such as the LIAR and CANYON algorithms, which are available and based on more data. Why not use one of them (CANYON may not work as it requires O₂, but LIARv2 certainly has T/S only selections)? Is there a significant difference?

On line 280 and elsewhere the term glider is used to refer to the Waveglider. That's relatively confusing as the term glider is generally applied to buoyancy driven profiling platforms. It would be more helpful just to use Waveglider.

Reviewer #2 (Remarks to the Author):

Review of "Storms drive outgassing of CO₂ in the subpolar Southern Ocean"

For Nature Communications.

Summary:

This work analyses the results of a novel data set – derived from the combination of a profiling subsurface glider and a surface wave glider – to give insight into the evolution of near surface carbon parameters and air-sea fluxes over a summer period in the subpolar Southern Ocean. Besides the very high latitude of these observations, capturing turbulence, hydrography and air-sea fluxes simultaneously is a powerful combination. Using these data the authors explore the drivers of the air-sea carbon flux on synoptic and record length (season) time scales. They identify two main mechanisms that control the air-sea flux – wind-driven lateral Ekman currents that advect high carbon waters from the south (low carbon waters from the north) and punctuated occurrences of entrainment of high carbon waters from below the mixed layer under extreme storm conditions (and in the early part of the record when the summer stratification is still weak). A key result is that a one-dimensional interpretation of these data is completely inadequate, and that lateral flows in the mixed layer are first order on the synoptic timescale. What I find remarkable is how much of the observed air-sea flux can be modelled using just the wind and MEAN lateral DIC gradient. In the very dynamic Southern Ocean, this seems remarkable, but must speak to the lateral space scales and times scales of the Ekman fluxes vs the ACC fronts. More could be said on this.

The authors then go on to show that the dominance of the Ekman mechanism around the Southern Ocean is largely mediated by the strength and location of lateral DIC fronts, which are associated with the jets of the ACC. More should be said about what maintains these lateral DIC gradients which underpin the Ekman mechanism (UCDW upwelling/ACC frontal dynamics), and whether the air sea fluxes associated with Ekman mechanism rectifies into a net flux on seasonal timescales or simply cancels out over time. This is a crucial point and not really clearly stated. Besides the challenges of aliasing in observations (which apply to nearly all air-sea flux measurements), there are other results that I would stress here – the power of this novel data set, punctuated entrainment events, dominance of the simple Ekman mechanism, what this means for the larger and seasonal timescale flux dynamics (besides the aliasing challenge).

Thus I recommend the paper for publication, but with revisions to more clearly state the major novel conclusions. In addition, there is repetition of material and arguments in the manuscript that could be removed to allow more of the supplementary material to be included. E.g. Figure S4 seems key, and possibly S1.

Minor comments:

Lines 29 – 32. The authors go from a sentence on anthropogenic CO₂ to the natural fluxes. It might be good to add a transition noting this shift – and contrasting their sizes?

Lines 31: "The upwelling drives COMPRISES THE large-scale surfacing of deep waters with high concentrations in dissolved inorganic carbon"

Figure 1a – shouldn't the background map be for the season of the experiment, not the annual mean?

Line 61 – this is a good point, but what do you show in this paper to address this question. This is not clear to me at all.

The plotted MLD does appear rather lacking in synoptic variability. I believe this results from the authors choice of an unusually large delta density of 0.03 – 3 times normal value of .01kg/m³. Thus they chose what could be more normally termed the base of the seasonal thermocline. Some comment should be made about this point, as the MLD they plot is not really a conventional density difference MLD, which is normally more sensitive to synoptic conditions and will be closer to the XLD. Thus the statement : "The MLD was not sensitive to variability of the wind..." should be qualified – maybe say 'depth of seasonal thermocline is not sensitive....'

There are 2 'entrainment events' – but only one shows increased salinity. Is it really entraining UCDW?

L 224 – note that this model ignores diffusion through MLD or lateral mixing.

Discussion – most of it is a summary and repeats material already presented. What are the important consequences of the results here?

Figure S9 – very neat. Put in main paper. Shows spatial gradients of $dDIC/dy$ is what sets synoptic variability of pCO_2 .

Question not addressed – what impacts does synoptic Ekman flow have on the net flux? Is this just a 'sloshing around' – that cancels out in space and time - uptake here, degassing there? Is this the result of the fast mixing in the atmosphere vs slow mixing in the ocean?

Reviewer #3 (Remarks to the Author):

Nicholson et al. report results from a novel experiment that paired a profiling glider and a wave glider carrying a pCO_2 system. These new observations indicate that short term (10 day) air-sea CO_2 fluxes can be dominated by Ekman transport and entrainment, both of which act to supply dissolved inorganic carbon to the mixed layer and enhance a positive air-sea gradient, which increases outgassing. During a 2-month experiment in the Atlantic sector of the Southern Ocean, they observed 2 short events of significant magnitude relative to the annual mean flux.

While the extrapolation of this synoptic scale variability and its impact on mean fluxes to the broader subpolar Southern Ocean is not without issue, the authors do a credible job of identifying the caveats in their approach.

The work is novel and will be of interest to a broad readership. I recommend the paper for publication – some minor comments are below.

Line 23: I think this should be the synoptic variability of the air-sea CO_2 gradient, or the air-sea CO_2 flux, not the 'synoptic variability of CO_2 '.

Line 29: please reword as '... total mean annual ocean uptake...'

Line 66: ΔpCO_2 is the gradient, not the flux

Line 81 (and throughout): please put the 'sea', 'atm' and later 'DIC' and 'SST' as superscripts, not subscripts.

Line 90: is 12 μatm really 'highly variable'?

Line 93: about 20? Is this referring to the 2x12 listed above?

Line 144 : model of who? List the author in reference 22 here please

Lines 276, 277: this seems to be one of the MOST interesting points of the paper – can you highlight it further?

Line 322: I am surprised that the biological component is so small in the summer season. This is briefly discussed later (Line 393) as a caveat, but might be worth mentioning here as well.

Line 406 – please include a section of the Alkalinity with the supplementary figure for DIC

Line 505: why not use the Takahashi et al., for the computation of alkalinity from T and S (Taro Takahashi, S.C. Sutherland, D.W. Chipman, J.G. Goddard, Cheng Ho, Timothy Newberger, Colm Sweeney, D.R. Munro, Climatological distributions of pH, pCO_2 , total CO_2 , alkalinity, and $CaCO_3$ saturation in the global surface ocean, and temporal changes at selected locations, Marine Chemistry, 164, 2014)?

Line 506: please include the equilibrium constants used, and estimates of the errors in the computed parameters (e.g., Orr, J. C., Epitalon, J.-M., Dickson, A. G., & Gattuso, J.-P. (2018). Routine uncertainty propagation for the marine carbon dioxide system. Marine Chemistry, 207,

84-107.)

REVIEWER COMMENTS

Reviewer #1 (Remarks to the Author):

The manuscript by Nicholson et al. is built on an impressive dataset that combines an array of novel observing technologies that have been applied in a very interesting study. The major focus of this study is a demonstration in the sub-of significant variability in surface ocean pCO₂ that is driven by storms. I.e., “air-sea gradient of CO₂ is dominated by synoptic storm-driven ocean variability” (lines 18/19). The data set is focused on two months of in situ pCO₂ measurements collected by a CO₂ sensor on a waveglider that operated during austral summer. I am not completely convinced of the major conclusions, however.

While this is quite a detailed analysis, I'm not sure that the basic premises are appropriate. The works was done near 54S, south of the Polar Front. This is a zone where the seasonal cycle of pCO₂ is driven by biological processes. The graphic pasted below shows a Hovmoller diagram of the annual cycle of pCO₂ in the South Atlantic, based on the climatology produced by the late Taro Takahashi (here made using ODV software and the data file downloaded from the ODV website). In the subtropics, the pCO₂ annual cycle is driven by temperature, with highest values in summer. South of the 40S, however, biological uptake of inorganic carbon dominates over the effects of temperature. The annual cycle inverts, relative to the subtropics, due to biological uptake of DIC during spring and summer that reduces pCO₂ more than the temperature effect on solubility increases pCO₂. This has been well recognized in numerous assessments. Yet, the analysis in the manuscript is focused on the balance of temperature effects driven by changes in CO₂ solubility (pCO₂-sst) and a pCO₂-DIC component that is “primarily caused by wind-driven DIC transport in the ocean”. The role of the major driving force, inorganic carbon uptake during photosynthesis, and the variability that this might drive and which dominates the seasonal cycle of pCO₂, is just not considered explicitly.

Given that biological uptake is the process that really drives the summer balance of CO₂ flux, I wouldn't think that (lines 319/321) “the sign of $\Delta p\text{CO}_2$ is dependent on synoptic (1-10 day) variability (approximately 20 μatm ; Fig. 1b) and thus key to constraining the mean FCO₂ over the summer season.” The variability that must be driven by biology is apparent in the data via their statements such as, “we observed several other synoptic-scale reversals in $\Delta p\text{CO}_2$ between uptake and outgassing events that were not explained by enhanced wind, mixing, and entrainment.” Biological uptake of carbon and reductions in pCO₂ are also likely

responsible for lags between wind forcing and pCO₂ changes (“events on 25/12 and 05/02 had slightly delayed observed responses”).

In summary, I believe that this is a great set of observations. The two events (28 Dec, 5 Jan) nicely show the effects of deepened mixing on CO₂ flux. On the other hand, Figure 2e doesn't seem to show that those two events would dominate the integrated flux over the observational period. As a result I wouldn't necessarily be convinced by the manuscripts conclusions.

We are grateful for the reviewer's comments, which led us to clarify and strengthen the findings of this study. In particular, we thank the reviewer for highlighting these potentially important considerations concerning the impact of biological driven variability in the subpolar region. We have carried out two further investigations which consider the role of biology on the scales that are critical to this study in this region. First, we consider the consequences of biology for the seasonal trend and then for synoptic variability in pCO_{2-DIC}. We find that indeed biology is important to consider as a driver of the seasonal trend, but we show that biology is a weak driver of synoptic variability compared to physical transport. Our result that physical transport drives the pCO₂ synoptic scale variability is thus strengthened. The underlying analysis is as follows:

1) Biology and seasonal variability of pCO_{2-DIC}. Firstly, we use maps of satellite derived net primary production (NPP) in the austral subpolar region (Fig S4a,b) as well as the annual mean and austral summer mean NPP estimated from several primary production models. This revealed that the subpolar region considered in this study has lower NPP relative to other zonal regions in the Southern Ocean. This is evident both during the period of sampling 2018 Dec - end of Feb 2019, as well as in annual mean. There are zonal exceptions to the low NPP in the region surrounding South America and near subpolar islands (Kerguelen). We have added this figure into supplementary:

Fig. S4.

Monthly mean net primary productivity (NPP, in units $\text{mg C m}^{-2} \text{d}^{-1}$) derived from satellite ocean color using three different primary models: the Carbon-based Productivity Model (CbPM⁷⁴), the Vertically Generalized Production Model (VGPM⁷⁵) and the Platt model⁷⁶. The mean of the three primary production models is shown in (a) averaged for 2018-2019 and (b) averaged over summer 2018-2019 coinciding with the deployment of the Gliders. (c) The time-series of satellite derived NPP using the three primary production model estimates are compared with the estimate derived by applying the Carbon-based Productivity Model⁷⁴ to optical measurements on the Slocum glider (dashed-line).

Secondly, we have also estimated *in situ* net primary production using the optical sensors on the buoyancy glider. The Slocum glider was fitted with a WETLabs ECO puck™ (BB2FI-470/700), thus measuring chlorophyll-a fluorescence (proxy for phytoplankton concentration) and two wavelengths of optical backscattering by particles, *bbp*(470) and *bbp*(700). The optics data were cleaned and processed following procedures recommended by *GliderTools* (Gregor et al. 2019). Backscatter, *bbp*(700), was converted to phytoplankton carbon following Behrenfeld et al (2005). Slocum glider observed NPP was estimated from the phytoplankton carbon, chlorophyll and collocated photosynthetically available radiation (PAR) from MODIS (the Slocum glider did not have a PAR sensor) following the Carbon-based Productivity Model (CbPM)(Westberry, et al. 2008) steps followed provided by Arteaga et al. (2020).

We agree with the reviewer that biological uptake likely plays a role on the seasonal-scale $\text{pCO}_{2\text{DIC}}$ depletion and we have revised the description of the seasonal trend in $\text{pCO}_{2\text{DIC}}$ in the results to reflect the analysis reported here. In particular, the decreasing trend of $\text{pCO}_{2\text{DIC}}$ (as observed in Figure 1f) can be largely explained by NPP that may range from 150-500 $\text{mg C m}^{-2} \text{d}^{-1}$

$\text{C m}^{-2} \text{ d}^{-1}$ (~13-40 $\text{mmol C m}^{-2} \text{ d}^{-1}$) on average (based on the various bio-optical estimates in Fig S4 above). Integrating this NPP over 56 days yields a carbon consumption of about 700 to 2300 mmol C m^{-2} . The estimated seasonal depletion of DIC in the mixed-layer ~1000 mmol C m^{-2} is within this range. In particular, we estimate the seasonal drawdown of DIC in the mixed layer to be approximately 1000 mmol C m^{-2} , which is estimated by multiplying the mean change in surface DIC of 10 mmol C m^{-3} inferred from the Waveglider observations by a crude estimate of the depth over which surface DIC is well mixed (100 m). In addition to the quite large uncertainty in NPP as reflected in the spread between algorithms, another caveat is that some unknown fraction of this NPP may reflect recycled production and not net DIC drawdown. **Note, however, that regardless of the driving mechanism this seasonal decrease in $\text{pCO}_{2\text{DIC}}$ is compensated by an increase in $\text{pCO}_{2\text{SST}}$ due to thermal warming, thus the summer seasonal trends of $\text{pCO}_{2\text{DIC}}$ and $\text{pCO}_{2\text{SST}}$ cancel out during our observational period (Fig 1f).**

In addition to the new Fig S4 and new description of the methods used to calculate NPP, the results section contains the following highly modified/new text to address the cause of the seasonal trend in $\text{pCO}_{2\text{DIC}}$:

“Meanwhile, the weakening trend in $\text{pCO}_{2\text{DIC}}$ is likely a consequence of a gradual decrease in DIC by approximately 10 mmol C m^{-3} (or equivalently 1000 mmol C m^{-2} assuming the top 100 m is mixed) over 2 months due to biological productivity. Consistent with this inferred DIC drawdown of 1000 mmol C m^{-2} , bio-optical estimates of net primary productivity from sensors on the Slocum glider and satellites range from approximately 13-40 $\text{mmol C m}^{-2} \text{ d}^{-1}$ or equivalently 700-2300 mmol C m^{-2} over the 56-day deployment (Fig. S4). Ocean advection and a net freshwater flux due to precipitation could also influence this trend in $\text{pCO}_{2\text{DIC}}$ significantly (Fig. S3b), but the relative contributions of advection and freshwater fluxes are not investigated in this study (refer to ³²). Regardless of the driving mechanisms, the seasonal trends in $\text{pCO}_{2\text{DIC}}$ and $\text{pCO}_{2\text{SST}}$ approximately compensate for each other, thus ΔpCO_2 fluctuates about its initial value near zero, and the sign of ΔpCO_2 changes on synoptic timescales during the 2-month glider deployment at this site.”

2) Biology and synoptic variability of $\text{pCO}_{2\text{DIC}}$. Now, we turn to the role of biology in the observed synoptic variability of $\text{pCO}_{2\text{DIC}}$. Importantly in terms of the study objectives, our glider-derived estimates of NPP also provide evidence that NPP was unable to explain the synoptic-scale variability in the DIC and thus in the $\text{pCO}_{2\text{DIC}}$ anomalies as observed at the location of the gliders. This is clearly evidenced by the magnitude of the short-term variability (e.g. daily variations of NPP_{int} approximately $\pm 10 \text{ mmol C m}^{-2}$), which is at least an order of magnitude less than the short-term (daily-synoptic) changes in DIC, approximately of the order $\pm 200 \text{ mmol C m}^{-2}$ (this is calculated by scaling DIC to the mixed-layer, such that $\text{DIC}_{\text{MLD}} = \text{DIC} \cdot \text{MLD}$, where surface DIC is estimated from the WG observations and the MLD is obtained from the profiling glider). This is illustrated by the new Fig S9 below which has been added into the supplementary material. However, it is important to note that, while the variability is an order of magnitude lower, it is not completely negligible and could have some minor impacts on the phasing of the $\text{pCO}_{2\text{DIC}}$ when the amplitude of pCO_2 variability is small as suggested by the reviewer.

Fig S9.

Synoptic anomalies (the residual after removing a 10-day rolling mean) of dissolved inorganic carbon (DIC) scaled by the mixed-layer depth (DIC_{MLD} in units mmol C m^{-2} , left axis), where $\text{DIC}_{\text{MLD}} = \text{DIC} \cdot \text{MLD}$. The MLD is estimated from the profiling glider. DIC is estimated from Wave Glider observed $p\text{CO}_2$ and derived Total Alkalinity⁷² using PyCO2SYS⁷³. DIC_{MLD} is compared to the synoptic anomalies in time integrated net primary production (NPP_{INT} in units mmol C m^{-2} , right axis; note the scale is an order of magnitude smaller than the left axis) derived from the glider as described in Fig. S4.

To address the synoptic variability driven by biology, we revised the text as follows:

*“The resulting time series of the estimated $p\text{CO}_{2-\text{DIC}}$ ’ computed using the model (Eq. 2) is compared to the Wave Glider observed $p\text{CO}_{2-\text{DIC}}$ ’ (Fig. 3d). Strikingly, the modelled $p\text{CO}_{2-\text{DIC}}$ ’ reproduces most of the observed synoptic variability in $p\text{CO}_{2-\text{DIC}}$ ’, accounting for 60% of the total observed variance in $p\text{CO}_{2-\text{DIC}}$ ’. However, there are some discrepancies between the estimated and the observed values, such as the phasing of the estimated variability is not always aligned with the observed $p\text{CO}_{2-\text{DIC}}$ ’ (e.g., events on 25/12 and 05/02 had slightly delayed observed responses). Likewise, the estimated magnitude of the response is sometimes underestimated (12/01) or overestimated (30/12). Nevertheless, we conclude that physical transport associated with Ekman advection and entrainment and encapsulated in the model is the dominant cause of the observed synoptic variability in $p\text{CO}_{2-\text{DIC}}$, $\Delta p\text{CO}_2$ and $p\text{CO}_{2\text{sea}}$. **All other transport and biological processes can explain at most 40% of the variance and are therefore subdominant. In particular, variability in biological sources and sinks of DIC may explain a small fraction of the synoptic variance in $p\text{CO}_{2-\text{DIC}}$, but the amplitude of synoptic variations in net primary productivity derived from in situ optical measurements are estimated to be an order of magnitude too weak to explain the observed synoptic variability in $p\text{CO}_{2-\text{DIC}}$ (Fig S9).**”*

In the Methods, we add important related caveats about biological processes when extrapolating:

“Particular caution must be exercised when using the results of the conceptual model in regions where the neglected factors may be more significant, either because the strength of one or more of the neglected factors is greater or because the variability driven by Ekman and entrainment is weaker. With regard to the neglected NCP, for example, NPP is much stronger and more variable than at $54^\circ\text{S}, 0^\circ\text{W}$ in some localised regions surrounding sub-polar islands and off the coast of South America (Fig. S4). ... ”

The reviewer notes: Given that biological uptake is the process that really drives the summer balance of CO₂ flux, I wouldn't think that (lines 319/321) "the sign of $\Delta p\text{CO}_2$ is dependent on synoptic (1-10 day) variability (approximately 20 μatm ; Fig. 1b) and thus key to constraining the mean F_{CO_2} over the summer season.

While this statement is correct for our observations, we think the reviewer must have interpreted the text more generically than we intended. We have added some revised text in the discussion of broader implications to clarify this:

"to a first approximation (and by definition in Eq. 3), oscillatory synoptic perturbations to Ekman advection are reversible and do not sum to impact the mixed layer DIC budget and F_{CO_2} on longer timescales. To this level of approximation, perturbations to Ekman advection only locally modulate the sign of $\Delta p\text{CO}_2$ where the seasonal drivers associated with thermal and non-thermal effects yield small mean $|\Delta p\text{CO}_2|$ relative to the synoptic perturbations, and thus the sign of $\Delta p\text{CO}_2$ is highly sensitive to the synoptic perturbations and varies on synoptic timescales. In addition, the observed mean $\Delta p\text{CO}_2$ is only sensitive to the synoptic perturbations in Ekman advection to the degree that the synoptic variance is also relatively large compared to the number of synoptic events or the duration of sampling (i.e., standard errors are large)."

Finally the reviewer notes: "On the other hand, Figure 2e doesn't seem to show that those two events would dominate the integrated flux over the observational period." We have rephrased the text to clarify:

"These two events are associated with two of the largest CO₂ outgassing peaks observed during the experiment (Fig. 2e) and shifted the mean sea-to-air F_{CO_2} from $-0.12 \text{ mol C m}^{-2} \text{ yr}^{-1}$ (i.e., the mean excluding these two events of positive F_{CO_2}) to $-0.06 \text{ mol C m}^{-2} \text{ yr}^{-1}$ (i.e., the mean of the full record in Fig. 2e including these two events)."

and:

"entrainment does cause rare large $p\text{CO}_{2-\text{DIC}}$ ' anomalies and it contributes substantially to the strongest outgassing fluxes F_{CO_2} observed on 28 December 2018 and 5 January 2019 that result from a synergistic combination of positive $p\text{CO}_{2-\text{DIC}}$ ' and $\Delta p\text{CO}_2$ due to Ekman advection [i.e., storm-driven meridional advection by the Ekman flow] and entrainment as well as large K_w from strong winds (Fig. 2d)."

See also our response to reviewer 2 for some additional discussion about the generic significance of synoptic Ekman advection and entrainment for slower-timescale dynamics, including the new Fig 5.

Other comments:

There needs to be more information on the pCO₂ sensor used in the paper. The one cited reference near the mention of the sensor on line 452/453 provides no information on the instrument.

We have added further description of the sensor to this paragraph below:

"The Liquid Robotics SV3 Wave Glider (WG) was fitted with an Airmar WX-200 Ultrasonic Weather Station mounted on a mast at 0.7 m above sea level and sampled wind speed and

direction at a rate of 1 Hz, averaged into 10-minute bins. The surface winds were corrected to a height of 10 m above sea level as in ⁶¹. The WG was equipped with a SeaBird Glider Payload CT-cell, measuring surface ocean temperature and conductivity at 1 Hz, averaged into 20 minutes bins. In addition, the WG was fitted with a VeGAS-pCO₂ (Versatile Glider, Atmospheric and Ship pCO₂ high Precision pCO₂ analyzers) measuring atmospheric and ocean pCO₂. The VeGAS-pCO₂ sensor is based on the well-established NDIR (Licor - Li-820) linked equilibrator units^{62,80} but with significant redesign to improve accuracy (<1 μatm), precision (< 1 μatm) through more effective drying and temperature control, equilibrator design and long term stability that also reduced the frequency of reference gas calibration from every sample to every 2 hours. The unit was installed and linked to the SV3-WG control unit which enables remote communication and to send real time data. These instruments have just recently been successfully assessed in the ICOS Ocean Thematic Centre instrument intercomparison study and those results will be published through ICOS.”

Alkalinity estimates based on Lee 2006. There are much more comprehensive tools, such as the LIAR and CANYON algorithms, which are available and based on more data. Why not use one of them (CANYON may not work as it requires O₂, but LIARv2 certainly has T/S only selections)? Is there a significant difference?

Thank you for raising this important issue. The strength of gap-filling / transfer methods such as LIARv02 and CANYON lies in the ocean interior away from the mixed layer where high frequency variability impacts on the magnitude of the uncertainties. Both methods also perform best with additional ancillary variables, especially oxygen, whose variability and uncoupling from CO₂ in the ML does not contribute significantly to reducing the estimated errors Carter et al., 2017; Bittig et al., 2018). Thus the reconstruction of AT and DIC in the mixed layer is still best served by using t, S which can be measured accurately at high frequency. When restricted to t,S in the mixed layer Lee et al, (2006) remains at least as good as either CANYON or LIARv2.

For reference, LIARv2 yields a TA of 2308 μmol/kg for at 54 S, 0 W, 0 m depth and S=34.1 g/kg, T=1.0 deg C, with an uncertainty of 9 μmol/kg, which is comparable to the ±8μmol/kg from Lee2006.

On line 280 and elsewhere the term glider is used to refer to the Waveglider. That's relatively confusing as the term glider is generally applied to buoyancy driven profiling platforms. It would be more helpful just to use Waveglider.

Noted. We have cleared up this distinction in the text and made sure there is no confusion with either platform - Wave Glider or profiling glider.

References included in above response:

Behrenfeld, MJ, PG Falkowski (1997), *Photosynthetic rates derived from satellite-based chlorophyll concentration*. Limnology and Oceanography Volume 42: 1-20.

Platt, Trevor, and Shubha Sathyendranath. "Oceanic Primary Production: Estimation by Remote Sensing at Local and Regional Scales." *Science*, vol. 241, no. 4873, 1988, pp. 1613–1620. JSTOR, www.jstor.org/stable/1702228. Accessed 5 Aug. 2021.

Westberry, T., M. J. Behrenfeld, D. A. Siegel, E. Boss (2008). *Carbon-based primary productivity modeling with vertically resolved photoacclimation*. GBC. <https://doi.org/10.1029/2007GB003078>

Gregor, L. *et al.* GliderTools: A Python Toolbox for Processing Underwater Glider Data. *Front. Mar. Sci.* **6**, 1–13 (2019).

Arteaga, L.A., Boss, E., Behrenfeld, M.J. *et al.* Seasonal modulation of phytoplankton biomass in the Southern Ocean. *Nat Commun* **11**, 5364 (2020). <https://doi.org/10.1038/s41467-020-19157-2>.

Behrenfeld, Boss, Siegel, Donald and Shea (2005). *Carbon-based ocean productivity and phytoplankton physiology from space*. GBC. <https://doi.org/10.1029/2004GB002299>.

Platt, T., and Sathyendranath, S. 1993. Estimators of primary production for interpretation of remotely-sensed data on ocean color. *Journal of Geophysical Research: Oceans*, **98**: 14561–14576.

Reviewer #2 (Remarks to the Author):

Review of “Storms drive outgassing of CO₂ in the subpolar Southern Ocean”

For Nature Communications.

Summary:

This work analyses the results of a novel data set – derived from the combination of a profiling subsurface glider and a surface wave glider – to give insight into the evolution of near surface carbon parameters and air-sea fluxes over a summer period in the subpolar Southern Ocean. Besides the very high latitude of these observations, capturing turbulence, hydrography and air-sea fluxes simultaneously is a powerful combination. Using these data the authors explore the drivers of the air-sea carbon flux on synoptic and record length (season) time scales. They identify two main mechanisms that control the air-sea flux – wind-driven lateral Ekman currents that advect high carbon waters from the south (low carbon waters from the north) and punctuated occurrences of entrainment of high carbon waters from below the mixed layer under extreme storm conditions (and in the early part of the record when the summer stratification is still weak). A key result is that a one-dimensional interpretation of these data is completely inadequate, and that lateral flows in the mixed layer are first order on the synoptic timescale. What I find remarkable is how much of the observed air-sea flux can be modelled using just the wind and MEAN lateral DIC gradient. In the very dynamic Southern Ocean, this seems remarkable, but must speak to the lateral space scales and times scales of the Ekman fluxes vs the ACC fronts. More could be said on this.

The authors then go on to show that the dominance of the Ekman mechanism around the Southern Ocean is largely mediated by the strength and location of lateral DIC fronts, which are associated with the jets of the ACC. More should be said about what maintains these

lateral DIC gradients which underpin the Ekman mechanism (UCDW upwelling/ACC frontal dynamics), and whether the air sea fluxes associated with Ekman mechanism rectifies into a net flux on seasonal timescales or simply cancels out over time. This is a crucial point and not really clearly stated. Besides the challenges of aliasing in observations (which apply to nearly all air-sea flux measurements), there are other results that I would stress here – the power of this novel data set, punctuated entrainment events, dominance of the simple Ekman mechanism, what this means for the larger and seasonal timescale flux dynamics (besides the aliasing challenge).

Thus I recommend the paper for publication, but with revisions to more clearly state the major novel conclusions. In addition, there is repetition of material and arguments in the manuscript that could be removed to allow more of the supplementary material to be included. E.g. Figure S4 seems key, and possibly S1.

We thank the reviewer for their insightful comments which have helped us to clarify and punctuate the important conclusions that have been raised by this study. In particular, the reviewer has highlighted some key discussion points which were overlooked in the first version of the manuscript. We have removed areas of repetition throughout the manuscript but particularly in the discussion section and we have also moved some of the supplementary figures into the main manuscript (Figure S9 has now been merged with Figure 4 and Figure S4 has now been merged with Figure 1). We also added Figure 5 to highlight the importance of these intermittent entrainment events on longer timescales.

The reviewer notes above: What I find remarkable is how much of the observed air-sea flux can be modelled using just the wind and MEAN lateral DIC gradient. In the very dynamic Southern Ocean, this seems remarkable, but must speak to the lateral space scales and times scales of the Ekman fluxes vs the ACC fronts. More could be said on this. We have added the following text in the discussion to expand more on this important point:

“The hypothesized dominance of Ekman advection of mean gradients as a driver of the synoptic variability of pCO_{2-DIC} has some somewhat surprising and important implications. First, it implies that time-mean gradients $\frac{\partial(\langle DIC, A_T \rangle)}{\partial y}$ are generally larger than anomalies $\frac{\partial(DIC, A_T)'}{\partial y}$ on the time scales relevant to the synoptic lateral advection (that is days to weeks). This result in turn implies that the large-scale (> 500 km) A_T and DIC fronts are fairly stable in time (e.g., as seen by the low meridional variability of the DIC and A_T observed in Fig. 3b). Drivers of large-scale variability such as changes in the large-scale circulation, biological productivity, and air-sea fluxes evidently do not cause substantial inter-annual or even seasonal deviations from the time-mean $\frac{\partial(\langle DIC, A_T \rangle)}{\partial y}$. In addition, mesoscale (< 500 km) DIC and A_T gradients are also weak relative to the large-scale time-mean $\frac{\partial(\langle DIC, A_T \rangle)}{\partial y}$. Stirring by mesoscale eddies is relatively ineffective at producing A_T and DIC anomalies (e.g., via frontogenesis, see ⁴⁶) compared to the mechanisms that are destroying the mesoscale gradients. Relatedly, the dominance of Ekman advection also implies that Ekman velocities dominate all other sources of synoptic velocity variability in the mixed layer, that is mesoscale and submesoscale turbulent velocities are relatively weak compared to Ekman velocities on synoptic timescales. A full explanation of these results and a broader evaluation of these hypotheses is beyond the scope of this work, but it is important to recognize that the explanatory power of Ekman advection of mean gradients and hence the accuracy of the

extrapolations in Fig. 4 depend on the relative weakness of both variability in meridional gradients of $p\text{CO}_{2\text{-DIC}}$ and non-Ekman synoptic velocities, as inferred from observations in the SE Atlantic multi-glider deployment. We have not investigated what sets the magnitude of these large-scale mean lateral gradients of $p\text{CO}_{2\text{-DIC}}$ (Fig 4d) or the mesoscale kinetic energy; however, these topics have been explored in other papers and are thought to be associated with the large-scale ocean fronts of the Antarctic Circumpolar Current (e.g., ^{47,48,49}).”

Furthermore the reviewer notes: More should be said about what maintains these lateral DIC gradients which underpin the Ekman mechanism (UCDW upwelling/ACC frontal dynamics), and whether the air sea fluxes associated with Ekman mechanism rectifies into a net flux on seasonal timescales or simply cancels out over time. This is a crucial point and not really clearly stated. Besides the challenges of aliasing in observations (which apply to nearly all air-sea flux measurements), there are other results that I would stress here – the power of this novel data set, punctuated entrainment events, dominance of the simple Ekman mechanism, what this means for the larger and seasonal timescale flux dynamics (besides the aliasing challenge).

We have added the following paragraphs to the discussion addressing the implications of 1) the dominance of the Ekman advection, and 2) the intermittent entrainment events, for the larger/seasonal timescale dynamics:

Implications of storm-driven synoptic variability for carbon dynamics on longer timescales

Another outstanding question is whether or not the storm-driven Ekman advection and entrainment have implications for the slower seasonal or interannual carbon dynamics of the Southern Ocean? Here, we address this question in two parts, first focusing on the Ekman advection and then entrainment.

Perhaps the most striking result derived from the glider deployment and the subsequent extrapolation across the subpolar Southern Ocean is that a simple Ekman advection of mean meridional gradients explains the majority of the synoptic variance in $p\text{CO}_{2\text{sea}}$ and $\Delta p\text{CO}_2$ (Fig. 3 and Fig. 4). However, to a first approximation (and by definition in Eq. 3), oscillatory synoptic perturbations to Ekman advection are reversible and do not sum to impact the mixed layer DIC budget and F_{CO_2} on longer timescales. To this level of approximation, perturbations to Ekman advection only locally modulate the sign of $\Delta p\text{CO}_2$ where the seasonal drivers associated with thermal and non-thermal effects yield small mean $|\Delta p\text{CO}_2|$ relative to the synoptic perturbations, and thus the sign of $\Delta p\text{CO}_2$ and the resulting F_{CO_2} are highly sensitive to the synoptic perturbations and vary on synoptic timescales. In addition, the observed mean $\Delta p\text{CO}_2$ is only sensitive to the synoptic perturbations in Ekman advection to the degree that the synoptic variance is also relatively large compared to the number of synoptic events or the duration of sampling (i.e., standard errors are large). However, oscillatory Ekman advection can be rectified in other subtle ways that are not captured at this level of approximation. For example, oscillatory advection of otherwise static ocean $p\text{CO}_{2\text{sea}}$ spatial gradients under a spatially and temporally variable atmosphere may modify the average F_{CO_2} over longer time intervals due to correlation between Ekman-driven $\Delta p\text{CO}_2$ anomalies and wind speed as well as the non-linear dependence of F_{CO_2} on wind speed. In addition, the combination of oscillatory Ekman advection and intermittent ocean mixing (e.g. during entrainment events) may induce lateral mixing via shear dispersion that irreversibly sums to impact the slower evolution of the mixed layer $p\text{CO}_{2\text{sea}}$ (e.g., ⁴⁹). However, it is beyond the scope of this manuscript to quantify their consequences for the large-scale dynamics of DIC and the air-sea CO_2 flux.

On the other hand, entrainment events, which have been shown to have a much smaller contribution to the synoptic variance than Ekman advection (Figs. 3, 4a,b), connect the subsurface carbon-rich UCDW to the mixed layer irreversibly. Thus, all entrainment events sum to impact the mixed layer DIC budget and hence F_{CO_2} on longer timescales (seasonal to interannual). The probability of sampling short storm-driven entrainment events like those observed by the multi-glider deployment with a 10 day (e.g., floats) or greater sampling period (e.g., ships) is very low and the response to entrainment is obscured by Ekman advection in any case. Thus, it is not currently possible to observationally quantify intermittent synoptic entrainment fluxes across the entire subpolar Southern Ocean as we do in the SE Atlantic with this data from paired gliders; coarser spatio-temporal sampling will alias this variability¹⁷. Hence, we use the model Eq. 2-4 to provide an estimate of the magnitude of the time-averaged synoptic entrainment flux across the subpolar Southern Ocean. Fig. 5 quantifies the annual mean entrainment flux of DIC (Eq. 4, see also Methods) and compares it (for perspective) with the magnitude of the climatological seasonal amplitude of F_{CO_2} (see also Fig. 1a,b). It shows that mean storm-driven entrainment flux (approximately $3.5 \text{ mol C m}^{-2} \text{ y}^{-1}$) is of the similar order of magnitude to the amplitude of the seasonal cycle in F_{CO_2} (approximately $2.1 \text{ mol C m}^{-2} \text{ y}^{-1}$, peak to trough) as well as the time-mean F_{CO_2} (Fig 1a; approximately $-0.1 \text{ mol C m}^{-2} \text{ y}^{-1}$). Hence, even small variations in the synoptic entrainment flux of DIC have a large impact on the mixed layer DIC budget relative to F_{CO_2} . If changes in synoptic entrainment go uncompensated by changes in other sources/sinks of mixed layer DIC such as biological export production (which is plausibly of the same order of magnitude and opposite sign; see Fig. S4) or other physical transport process, they will drive changes in DIC and modified F_{CO_2} . Consideration of the spatial structure of synoptic entrainment integrated over a year in Fig. 5 shows that entrainment exhibits substantial spatial variability and is particularly strong in the South Atlantic where the MLD_{max} is comparatively shallow relative to the Pacific basin (²² their Fig 10c) and storms are more frequent and stronger.¹³ The spatial variability of the synoptic entrainment highlights the variable circumpolar implications of the observation reported above that storm-driven entrainment is sensitive to the winter MLD maximum (which sets the depth of UCDW reservoir) (Figs. 1b,c and 2b) and to stabilizing buoyancy forcing (which prevents storm-driven vertical mixing from reaching the UCDW reservoir during later summer months, e.g., in Fig 2).

Although a complete analysis of the mixed layer DIC budget is beyond the scope of this paper, the results in Fig. 5 indicate that it is possible that storm driven entrainment cumulatively impacts the mean $pCO_{2,sea}$ and CO_2 flux and influences interannual²⁸ and spatial variability of CO_2 outgassing (Fig. 1 and Fig. 5) through the interactions between annual changes in storm-characteristics⁵¹, the seasonal cycle of the mixed layer, and variations to the depth of the winter MLD maximum⁵². We leave tests of these hypotheses to future work, but the success of the conceptual model (Eq. 2-4) in describing observations in the Atlantic sector (Fig. 3) coupled with the spatially variable and significant implications of storms for the mixed layer DIC budget on a range of timescales (Figs. 4-5) motivate future field experiments and comprehensive modelling that can accurately quantify and predict the physical-carbon dynamics of the ocean mixed layer down to synoptic timescales more broadly around the subpolar Southern Ocean.

Fig. 5. Storm-driven entrainment has the same order-of-magnitude effect on the mixed layer DIC budget than the air-sea CO₂ flux across the subpolar Southern Ocean. For perspective, we plot (b) the climatological annual mean entrainment flux (Z_{ent} , see Eq. 4 and refer to Methods) of Dissolved Inorganic Carbon (DIC, $\text{mol m}^{-2} \text{yr}^{-1}$) in comparison with (a) the climatological seasonal amplitude of the seasonal cycle in the CO₂ flux (F_{CO_2} , $\text{mol m}^{-2} \text{yr}^{-1}$) from CSIR-ML6. Both (a) and (b) are estimated over the period 2005 – 2019. Both (a) and (b) may be compared with the time-mean F_{CO_2} in Fig. 1a.

Minor comments:

Lines 29 – 32. The authors go from a sentence on anthropogenic CO₂ to the natural fluxes. It might be good to add a transition noting this shift – and contrasting their sizes?

Noted. The revised text now reads:

“The Southern Ocean is a key component of the Earth’s carbon budget. It accounts for 40-50% of the total mean annual ocean uptake of anthropogenic CO₂ (~1 Pg C/yr)¹⁻⁴. In addition, a weakening of the annual mean outgassing of natural CO₂ from the Southern Ocean in the 1990s and the subsequent decrease in outgassing in the 2000s (resulting in a ~0.5 Pg C/yr reinvigoration of ocean uptake) showed that the global ocean carbon budget is sensitive to variability in the Southern Ocean⁵⁻¹⁰”

Lines 31: “The upwelling drives COMPRISES THE large-scale surfacing of deep waters with high concentrations in dissolved inorganic carbon”

Revised as suggested.

Figure 1a – shouldn’t the background map be for the season of the experiment, not the annual mean?

Noted, we have changed Figure 1 to now include the summer mean during the experiment in addition to the annual mean. The annual mean is used to define the subpolar region and thus, we have chosen to keep it in the MS. It is also referenced in comparison with the entrainment fluxes.

Fig. 1. Observed temporal variability of $\Delta p\text{CO}_2$ in the outgassing domain of the Southern Ocean. (a) The annual mean (2005 - 2019) net air-sea CO_2 flux (F_{CO_2} , $\text{mol C m}^{-2} \text{yr}^{-1}$) from the CSIR-ML6 product¹. Overlaid is the annual maximum sea ice concentration from NCEP-DOE AMIP-II Reanalysis 2³³ before deployment 18th December 2018. The deployment labelled “Gliders” comprises a Wave Glider and Slocum glider marked by a black dot. The subpolar outgassing region considered here is between the climatological sea ice-edge maximum and the extent of the zonal band of maximum outgassing for 2005 - 2019 determined by the 0 contour of the CO_2 flux in winter (June - August), shown by black contours. (b) the same as (a) except F_{CO_2} deployment period (Dec - Feb 2019). A meridional section of (c) Dissolved Inorganic Carbon ($\mu\text{mol kg}^{-1}$) and (d) Total Alkalinity ($\mu\text{mol kg}^{-1}$) along the Good Hope Line (AX25) during 2016 from GLODAPv2.2020⁴¹. The blue contours (27.3 and 27.8 kg m^{-3} isopycnals) are the upper and lower bounds of the Upper Circumpolar Deep Water (UCDW). White contour is the mixed layer depth (MLD). (e) Wave Glider observed $\Delta p\text{CO}_2$ ($p\text{CO}_{2\text{sea}} - p\text{CO}_{2\text{atm}}$) in μatm . Grey bars highlight the central part of a storm passage defined using the 25th sea level pressure and 75th wind speed percentiles. (f) Decomposition of $p\text{CO}_{2\text{sea}}$ into its thermal ($p\text{CO}_{2\text{-SST}}$) and non-thermal ($p\text{CO}_{2\text{-DIC}}$) drivers. The thick lines represent the cumulative contribution of each process to the observed changes in $\Delta p\text{CO}_2$ relative to the start of deployment (time = 0). The thin lines show the 10-day rolling mean. Time is given as dd\mm of 2018 and 2019.

Line 61 –“There remains a significant gap in the understanding of the mechanisms that drive variability on synoptic time scales and how this synoptic variability rectifies on the seasonal cycle and mean of CO₂ fluxes” this is a good point, but what do you show in this paper to address this question. This is not clear to me at all.

We revised the text and added figure 5 to make the rectification effects clearer. Here in the intro, we added a sentence as follows

“There remains a significant gap in the understanding of the mechanisms that drive variability on synoptic time scales and how this synoptic variability rectifies on the seasonal cycle and mean of CO₂ fluxes. We explain how storms influence through mixed layer physics (advection and mixing) the direction and magnitude of the air-sea CO₂ gradient ($\Delta p\text{CO}_2$) and flux (F_{CO_2}) over the duration of the experiment. Then, using a conceptual ocean mixed layer model that captures the observed synoptic variability of $\Delta p\text{CO}_2$ in the observations, we quantify the synoptic variability around the entire subpolar Southern Ocean and discuss the implications of synoptic processes for the mixed layer carbon budget and F_{CO_2} on longer timescales.”

In the results, we revised some text to emphasize that two storm events are “associated with two of the largest CO₂ outgassing peaks observed during the experiment (Fig. 2e) and shifted the mean sea-to-air F_{CO_2} by 50% from $-0.12 \text{ mol C m}^{-2} \text{ yr}^{-1}$ (i.e., the mean excluding these two events of positive F_{CO_2}) to $-0.06 \text{ mol C m}^{-2} \text{ yr}^{-1}$ (i.e., the mean of the full record in Fig. 2e including these two events).”

And we substantially added to the discussion including a separate subsection and figure 5 on this topic (see our response to your major comment on this issue).

The plotted MLD does appear rather lacking in synoptic variability. I believe this results from the authors choice of an unusually large delta density of 0.03 – 3 times normal value of .01kg m⁻³. Thus they chose what could be more normally termed the base of the seasonal thermocline. Some comment should be made about this point, as the MLD they plot is not really a conventional density difference MLD, which is normally more sensitive to synoptic conditions and will be closer to the XLD. Thus the statement : “The MLD was not sensitive to variability of the wind...” should we qualified – maybe say ‘depth of seasonal thermocline is not sensitive....’

To note, 0.03kg/m³ threshold for mixed layers is widely used in the literature for computing mixed-layer depths in the Southern Ocean (e.g., De Boyer Montegut et al. 2004; Dong et al. 2008). The threshold of 0.01 kg/m³ has been reported to yield too shallow mixed-layer depths (Montegut et al. 2004, Dong et al. 2008).

For example, see excerpt from the abstract of De Boyer Montegut et al. 2004:

“The criterion selected is a threshold value of temperature or density from a near-surface value at 10 m depth ($\Delta T = 0.2^\circ\text{C}$ or $\Delta\sigma\theta = 0.03 \text{ kg m}^{-3}$)”

From Dong et al. 2008:

“Smaller net difference values of DT (0.1°C) and Dr (0.01 kg m^3) result in shallower mixed layers, but the differences from MLD based on DT (0.2°C) and Dr (0.03 kg m^3) are mostly within 20 m. Those smaller net difference values are more likely affected by anomalous spikes and small perturbations in the profiles.”

Also, note that to quote Brainerd and Gregg “The mixed layer is the envelope of maximum depths reached by the mixing layer on timescales of a day or more and is the zone that has been mixed in the recent past. It generally corresponds to the zone above the top of the

seasonal pycnocline.” Thus, we stick with our definition of the MLD and merely add a note that our results are consistent with expectations and cite Brainerd and Gregg:

The MLD was not directly sensitive to variability of the wind ($r^2 = 0$, Fig. 2a-b) and was thus distinctly different from the XLD. This is consistent with our conventional density threshold definition of the MLD (refer to Materials and Methods), which is not expected to vary with the XLD on synoptic or shorter timescales.³⁶

There are 2 ‘entrainment events’ – but only one shows increased salinity. Is it really entraining UCDW?

Yes, using the glider Absolute Salinity (Fig 2c), we have calculated the magnitude of entrainment of Absolute Salinity when the XLD > MLDmax, see the Figure below.

L 224 – note that this model ignores diffusion through MLD or lateral mixing.

We clarified this and more:

“It may be noted that this model omits biological sources and sinks as well as many transport processes, including diffusion through MLD, lateral mixing, and several lateral and vertical advective processes, all of which turn out to be less significant than Ekman advection in driving the observed synoptic variability of pCO_{2-DIC} .”

Discussion – most of it is a summary and repeats material already presented. What is the important consequences of the results here?

We have eliminated summaries of observational results in the discussion. And, we more explicitly address the important consequences of the observations via extrapolations in Fig 4-5 and related text. See our response to your major comments.

Figure S9 – very neat. Put in main paper. Shows spatial gradients of $dDIC/dy$ is what sets synoptic variability of PCO_2 .

The reviewer raises an important point. We have included three panels (a), (c) and (d) from Figure S9, which explains what sets the spatial distribution of synoptic variability in pCO_{2-DIC} , by adapting Figure 4 in the MS. The updated Figure 4:

Fig 4: The spatial distribution of synoptic variance of $p\text{CO}_{2-\text{DIC}}$ in the subpolar Southern Ocean. (a) Modelled 7-day $p\text{CO}_{2-\text{DIC}}$ variance (μatm^2) computed and averaged for 2019 (color bar). Overlaid are hexagons of co-located 7-day $p\text{CO}_{2-\text{DIC}}$ variance (μatm^2) as observed from a Saildrone that circumnavigated Antarctica in 2019²⁸. The spatial correlation r^2 and associated p-value of estimated versus observed are indicated. Hexagons outside of the subpolar domain are displayed with transparency and are excluded from the statistics. (b) the relative contribution of Ekman (%) to the synoptic $p\text{CO}_{2-\text{DIC}}$ variability shown in (a). (c) shows the modelled estimated 7-day $p\text{CO}_{2-\text{DIC}}$ variance as in (a), instead computed using a spatially uniform gradient of A_T and DIC (μatm^2) in Eq. 3. Thus, comparing (a) with (c) shows that the spatial variability of $p\text{CO}_{2-\text{DIC}}$ is not driven by spatial variability in wind, but rather the spatial variability driven by spatially diverse meridional gradients of A_T and DIC and thus $p\text{CO}_{2-\text{DIC}}$ ($\mu\text{atm m}^{-1}$). This is further evidenced when comparing (a) with (d) The meridional gradients of $p\text{CO}_{2-\text{DIC}}$ ($\mu\text{atm m}^{-1}$). Black contours on all panels show the location of the climatological sea ice-edge maximum and the outgassing maximum for 2005 - 2019, as in Fig. 1.

Question not addressed – what impacts does synoptic Ekman flow have on the net flux? Is this just a ‘sloshing around’ – that cancels out in space and time - uptake here, degassing there? Is this the result of the fast mixing in the atmosphere vs slow mixing in the ocean?

See our response to the second major comment, specifically the paragraph starting “*Perhaps the most striking...*”

References cited above:

de Boyer Montégut, C., Madec, G., Fischer, A.S., Lazar, A. and Iudicone, D., 2004. Mixed layer depth over the global ocean: An examination of profile data and a profile-based climatology. *Journal of Geophysical Research: Oceans*, 109(C12)

Dong, S., Sprintall, J., Gille, S.T. and Talley, L., 2008. Southern Ocean mixed-layer depth from Argo float profiles. *Journal of Geophysical Research: Oceans*, 113(C6).

Hoskins, B.J., 1982. The mathematical theory of frontogenesis. *Annual review of fluid mechanics*, 14(1), pp.131-151

McWilliams, J.C., 2021. Oceanic frontogenesis. *Annual Review of Marine Science*, 13, pp.227-253

Reviewer #3 (Remarks to the Author):

Nicholson et al. report results from a novel experiment that paired a profiling glider and a wave glider carrying a pCO₂ system. These new observations indicate that short term (10 day) air-sea CO₂ fluxes can be dominated by Ekman transport and entrainment, both of which act to supply dissolved inorganic carbon to the mixed layer and enhance a positive air-sea gradient, which increases outgassing. During a 2-month experiment in the Atlantic sector of the Southern Ocean, they observed 2 short events of significant magnitude relative to the annual mean flux.

While the extrapolation of this synoptic scale variability and its impact on mean fluxes to the broader subpolar Southern Ocean is not without issue, the authors do a credible job of identifying the caveats in their approach.

The work is novel and will be of interest to a broad readership. I recommend the paper for publication – some minor comments are below.

We thank the reviewer for their time spent on this manuscript and for their comments listed below. We have addressed the comments below (responses in blue text):

Line 23: I think this should be the synoptic variability of the air-sea CO₂ gradient, or the air-sea CO₂ flux, not the 'synoptic variability of CO₂'.

Noted, changed to "air-sea CO₂ gradient".

Line 29: please reword as '... total mean annual ocean uptak...'

Revised as suggested.

Line 66: DeltapCO₂ is the gradient, not the flux

Noted, this sentence now reads:

"We explain how storms influence through ocean mixed layer physics (advection and mixing) the direction and magnitude of the air-sea CO₂ gradient ($\Delta p\text{CO}_2$) and flux (F_{CO_2}) over the duration of the experiment."

Line 81 (and throughout): please put the 'sea', 'atm' and later 'DIC' and 'SST' as superscripts, not subscripts.

We have not made this stylistic change, because it would require redrawing all the figures for consistency and does not impact the interpretation of the results.

Line 90: is 12 μatm really 'highly variable'?

We would say yes, but given the reviewer finds this questionable, we have deleted the unclear "highly variable" in favor of a purely quantitative description: *"The observed $p\text{CO}_{2\text{sea}}$ and hence $\Delta p\text{CO}_2$ varied by approximately $\pm 10 \mu\text{atm}$, as F_{CO_2} oscillated between uptake and outgassing on synoptic timescales (1 - 10 days) (Fig. 1e). Several of the outgassing events coincided with the passage of storms (Fig. 1e - compare grey and red shaded areas). To put these results into perspective, the synoptic variability of $\Delta p\text{CO}_2$ (about $20 \mu\text{atm}$ from peak to trough) is similar in magnitude to the seasonal amplitude of $\Delta p\text{CO}_2$ and $p\text{CO}_{2\text{sea}}$ for the subpolar Southern Ocean^{10,32}."*

For reference, float observed changes in the seasonal cycle in this subpolar region (refer to Figure 3 of Williams et al. 2018, float 9096, and refer to their Table 2 for Antarctic Southern Zone) show seasonal change (winter - summer extreme) of $p\text{CO}_2$ of approximately up to $36 \pm 7 \mu\text{atm}$ over several months. Thus, in this seasonal context, $p\text{CO}_2$ variability of the order of up to $20 \mu\text{atm}$ (trough to peak of a synoptic cycle) that occurs every < 10 days is highly variable.

Line 93: about 20? Is this referring to the 2x12 listed above?

Yes, correct. To make this connection clearer we have updated the text above to:

*"varying between **approximately $\pm 10 \mu\text{atm}$** " and clarified that 20 is **"from peak to trough"***

Line 144 : model of who? List the author in reference 22 here please

Changed to: This is consistent with the model of Whitt et al. (2019)²²

Lines 276, 277: this seems to be one of the MOST interesting points of the paper – can you highlight it further?

We added further discussion around the importance of lateral gradients in DIC and AT for determining the magnitude of the synoptic variance in the Discussion:

"In agreement with the in situ observations from the multi-glider deployment (see Fig. 3), lateral Ekman advection over vertical entrainment is the dominant physical driver of the estimated high-frequency variability explaining about 92% of the model estimated variance of $p\text{CO}_{2\text{-DIC}}$ across the subpolar Southern Ocean (Fig. 4 b). Importantly, the spatial variation in the meridional gradients of $p\text{CO}_{2\text{-DIC}}$ explained most of the spatial variation in high-frequency temporal variability of $p\text{CO}_{2\text{-DIC}}$ (Fig. 4 a, c, d). This is consistent with Ekman advection scaling with the zonal wind stress and meridional gradients of DIC and Total Alkalinity $\frac{\partial(\text{DIC}, A_T)}{\partial y}$, because the spatial variability of the high-frequency wind is relatively uniform across this region (synoptic atmospheric variability occurs across large spatial scales of order 1000 km, refer Fig. 4 c).

“The hypothesized dominance of Ekman advection of mean gradients as a driver of the synoptic variability of $p\text{CO}_{2\text{-DIC}}$ has some somewhat surprising and important implications. First, it implies that time-mean gradients $\frac{\partial(\langle \text{DIC}, A_T \rangle)}{\partial y}$ are generally larger than anomalies $\frac{\partial(\text{DIC}, A_T)}{\partial y}$ on the time scales relevant to the synoptic lateral advection (that is days to weeks). This result in turn implies that the large-scale (> 500 km) A_T and DIC fronts are fairly stable in time (e.g., as seen by the low meridional variability of the DIC and A_T observed in Fig. 3b). Drivers of large-scale variability such as changes in the large-scale circulation, biological productivity, and air-sea fluxes evidently do not cause substantial inter-annual or even seasonal deviations from the time-mean $\frac{\partial(\langle \text{DIC}, A_T \rangle)}{\partial y}$. In addition, mesoscale (< 500 km) DIC and A_T gradients are also weak relative to the large-scale time-mean $\frac{\partial(\langle \text{DIC}, A_T \rangle)}{\partial y}$. Stirring by mesoscale eddies is relatively ineffective at producing A_T and DIC anomalies (e.g., via frontogenesis, see ⁴⁶) compared to the mechanisms that are destroying the mesoscale gradients. Relatedly, the dominance of Ekman advection also implies that Ekman velocities dominate all other sources of synoptic velocity variability in the mixed layer, that is mesoscale and submesoscale turbulent velocities are relatively weak compared to Ekman velocities on synoptic timescales. A full explanation of these results and a broader evaluation of these hypotheses is beyond the scope of this work, but it is important to recognize that the explanatory power of Ekman advection of mean gradients and hence the accuracy of the extrapolations in Fig. 4 depend on the relative weakness of both variability in meridional gradients of $p\text{CO}_{2\text{-DIC}}$ and non-Ekman synoptic velocities, as inferred from observations in the SE Atlantic multi-glider deployment. We have not investigated what sets the magnitude of these large-scale mean lateral gradients of $p\text{CO}_{2\text{-DIC}}$ (Fig 4d) or the mesoscale kinetic energy; however, these topics have been explored in other papers and are thought to be associated with the large-scale ocean fronts of the Antarctic Circumpolar Current (e.g., ^{47,48,49}). ”

In addition, we have modified Figure 4, to include three panels (a), (c) and (d) from former Figure S9, which explains what sets the spatial distribution of synoptic variability in $p\text{CO}_{2\text{-DIC}}$, now Fig 4 b,c,d in the main manuscript. It shows that gradients of DIC and T_a are what set the synoptic variability:

Fig. 4. The spatial distribution of synoptic variance of $p\text{CO}_{2-\text{DIC}}$ in the subpolar Southern Ocean. (a) Modelled 7-day $p\text{CO}_{2-\text{DIC}}$ variance (μatm^2) computed and averaged for 2019 (color bar). Overlaid are hexagons of co-located 7-day $p\text{CO}_{2-\text{DIC}}$ variance (μatm^2) as observed from a Sailability that circumnavigated Antarctica in 2019²⁸. The spatial correlation r^2 and associated p-value of estimated versus observed are indicated. Hexagons outside of the subpolar domain are displayed with transparency and are excluded from the statistics. **(b)** the relative contribution of Ekman (%) to the synoptic $p\text{CO}_{2-\text{DIC}}$ variability shown in (a). **(c)** shows the modelled 7-day $p\text{CO}_{2-\text{DIC}}$ variance as in (a), instead computed using a spatially uniform gradient of A_T and DIC (μatm^2) in Eq. 3. Thus, comparing (a) with (c) shows that the spatial variability of $p\text{CO}_{2-\text{DIC}}$ is not driven by spatial variability in wind, but rather the spatial variability driven by spatially diverse meridional gradients of A_T and DIC and thus $p\text{CO}_{2-\text{DIC}}$ ($\mu\text{atm m}^{-1}$). This is further evidenced when comparing (a) with **(d)** the meridional gradients of $p\text{CO}_{2-\text{DIC}}$ ($\mu\text{atm m}^{-1}$). Black contours on all panels show the location of the climatological sea ice-edge maximum and the outgassing maximum for 2005 - 2019, as in Fig. 1.

Line 322: I am surprised that the biological component is so small in the summer season. This is briefly discussed later (Line 393) as a caveat, but might be worth mentioning here as well.

See our response to reviewer 1 about the biological component. We have included estimates of NPP in the paper now and two new supplementary figures and some text. With regard to the observations in the SE Atlantic:

“variability in biological sources and sinks of DIC may explain a small fraction of the synoptic variance in $p\text{CO}_{2-\text{DIC}}$ but the amplitude of synoptic variations in net primary productivity derived from optical measurements are estimated to be an order of magnitude too weak to explain the observed synoptic variability in $p\text{CO}_{2-\text{DIC}}$ (Fig S9).”

With regard to the caveats in the discussion, we revised that to read:

“Particular caution must be exercised when using the results of the conceptual model in regions where the neglected factors may be more significant, either because the strength of one or more of the neglected factors is greater or because the variability driven by Ekman and entrainment is weaker. With regard to the neglected NCP, for example, NPP is much stronger and more variable than at $54^{\circ}\text{S}, 0^{\circ}\text{W}$ in some localised regions surrounding sub-polar islands and off the coast of South America (Fig. S4). Conversely, physical DIC variability due to Ekman advection is substantially weaker in areas where meridional DIC gradients are weak (Fig 4d).”

Line 406 – please include a section of the Alkalinity with the supplementary figure for DIC

We have included the sections of DIC (from the supplementary figure) and Alkalinity section mentioned above into the main manuscript as part of Figure 1, see below:

Fig. 1. Observed temporal variability of $\Delta p\text{CO}_2$ in the outgassing domain of the Southern Ocean. (a) The annual mean (2005 - 2019) net air-sea CO_2 flux (F_{CO_2} , $\text{mol C m}^{-2} \text{ yr}^{-1}$) from

CSIR-ML6¹. Overlaid is sea ice concentration maximum from NCEP-DOE AMIP-II Reanalysis 2³³. The deployment location of the robotic platforms (labelled “Gliders” comprises a Wave Glider and Slocum glider) marked by the black dot. The subpolar outgassing region considered here is between the climatological sea ice-edge maximum and the extent of the zonal band of maximum outgassing for 2005 - 2019 determined by the 0 contour of the CO₂ flux in winter (June - August), shown by black contours. (b) the same as (a) except F_{CO2} is averaged over the deployment period (Dec - Feb 2019) **An example meridional section of (c) Dissolved Inorganic Carbon (μmol kg⁻¹) and (d) Total Alkalinity (μmol kg⁻¹) along the Good Hope Line (AX25) during 2016 from GLODAPv2.2020⁴¹. The blue contours (27.3 and 27.8 kg m⁻³ isopycnals) are the upper and lower bounds of the Upper Circumpolar Deep Water (UCDW). White contour is the mixed layer depth (MLD).** (e) Wave Glider observed ΔpCO₂ (pCO_{2,sea} - pCO_{2,atm}) in μatm. Grey bars highlight the central part of a storm passage defined using the 25th sea level pressure and 75th wind speed percentiles (Fig S2). (f) Decomposition of pCO_{2,sea} into its thermal (pCO_{2-SST}) and non-thermal (pCO_{2-DIC}) drivers. The thick lines represent the cumulative contribution of each process to the observed changes in ΔpCO₂ relative to the start of deployment (time = 0). The thin lines show the 10-day rolling mean. Time is given as dd\mm of 2018 and 2019.

Line 505: why not use the Takahashi et al., for the computation of alkalinity from T and S (Taro Takahashi, S.C. Sutherland, D.W. Chipman, J.G. Goddard, Cheng Ho, Timothy Newberger, Colm Sweeney, D.R. Munro, Climatological distributions of pH, pCO₂, total CO₂, alkalinity, and CaCO₃ saturation in the global surface ocean, and temporal changes at selected locations, Marine Chemistry, 164, 2014)?

The Takahashi et al., 2014 approach suggested by the reviewer is based on the climatological relationship between salinity and alkalinity and accounts for secondary net community production effects on alkalinity using nitrate. In contrast, Lee et al. calculate alkalinity using quadratic fits to salinity and temperature; the temperature is a proxy to account for net community production among other effects. Lee et al. also construct their fits in different regions than Takahashi. The differences in alkalinity between Takahashi and Lee are shown to be relatively small <5 μmol/kg in our study area during Austral summer by Takahashi in their Fig 11. We think that for the specific needs of our study the Lee2006 method based on both t,S remains the best approach for our use case with the least number of underlying assumptions.

Line 506: please include the equilibrium constants used, and estimates of the errors in the computed parameters (e.g., Orr, J. C., Epitalon, J.-M., Dickson, A. G., & Gattuso, J.-P. (2018). Routine uncertainty propagation for the marine carbon dioxide system. Marine Chemistry, 207, 84–107.)

We included mention of the following constants: K1 and K2 from Merbach et al., 1973 refitted by Dickson and Millero, 1987, KHSO₄ Dickson, BT Uppstrom, 1974.

We also note that the estimated propagated error on the derived DIC is dominated by and comparable in magnitude to the uncertainty in derived TA (Orr et al., 2018), which is roughly eTA~ eDIC~ 5-10 μmolKg⁻¹SW (Lee et al. 2006; Orr et al., 2018). Comparable uncertainties are reported by Takahashi et al. (2014) and Carter et al. (2018) (LIARv2). However, it is important to note that the uncertainty in the individual values of DIC and TA does not propagate to pCO_{2-DIC} in the way of Orr et al. (2018) (e.g., eDIC*dpCO₂/dDIC ~ 15-30 uatm),

because $p\text{CO}_{2\text{-DIC}}$ is calculated as a residual from direct $p\text{CO}_2$ measurements on the glider, using direct temperature measurements to subtract off the variations due to temperature. That is, the uncertainty in $p\text{CO}_{2\text{-DIC}}$ is still only $\sim 2\text{uatm}$ because the uncertainty in the observation of temperature, used to derive the $p\text{CO}_{2\text{-DIC}}$ residual, is insignificant in the error propagation. Similarly, individual uncertainties in DIC or TA of $\sim 10\text{ umol/kg}$ would introduce uncertainties of 10% or $0.3\text{ uatm/umol/kgSW}$ in gradients $dp\text{CO}_2/d\text{TA}$ and $dp\text{CO}_2/d\text{DIC}$ and hence the modeled $p\text{CO}_{2\text{-DIC}}$. However, because of the constraint from directly observed $p\text{CO}_{2\text{sea}}$ (2 uatm), these uncertainties are negligibly small because changes in the gradients are much smaller along a line of constant $p\text{CO}_2$ in DIC/TA parameter space. We feel this is too much detail for the manuscript because the results don't really rely much on the precise DIC or TA values.

References included in above response:

Williams, N. L. *et al.* Assessment of the Carbonate Chemistry Seasonal Cycles in the Southern Ocean From Persistent Observational Platforms. *J. Geophys. Res. Ocean.* **123**, 4833–4852 (2018)

REVIEWERS' COMMENTS

Reviewer #1 (Remarks to the Author):

As I noted in my first review, this is an impressive data set. There is some nice analysis. Those results are marred by some overly broad conclusions. As an example, the abstract states "we show that the air-sea gradient of CO₂ is dominated by synoptic storm-driven ocean variability". That is really not correct. As I noted in my first review, the air-sea gradient is dominated by the effects of biological productivity that create a negative air-sea flux. The storm-driven variability is modulating the flux, but it is not dominating the flux. If it were not for the overwhelming signal due to biological uptake of CO₂, the air-sea flux would be positive due to the effects of seasonal heating, as is the case further north. The authors recognize this, albeit indirectly, with the statement they have added (line 191 and further in the Rebuttal document), "These two events are associated with two of the largest CO₂ outgassing peaks observed during the experiment (Fig. 2e) and shifted the mean sea-to-air FCO₂ from -0.12 mol C m⁻² yr⁻¹ (i.e., the mean excluding these two events of positive FCO₂) to -0.06 mol C m⁻² yr⁻¹ (i.e., the mean of the full record in Fig. 2e including these two events)." It is only through biological uptake that the flux is set to a negative value. It is certainly not surprising that storms modulate the air-sea flux, but the storms do not dominate the flux. The authors are overstating the impact of a nice study that illustrates the mechanisms by which storms can affect the air-sea flux.

Another example of an overly broad statement (line 434) is "The probability of sampling short storm driven entrainment events like those observed by the multi-glider deployment with a 10 day (e.g., floats) or greater sampling period (e.g., ships) is very low and the response to entrainment is obscured by Ekman advection in any case. Thus, it is not currently possible to observationally quantify intermittent synoptic entrainment fluxes across the entire subpolar Southern Ocean as we do in the SE Atlantic with this data from paired gliders; coarser spatio-temporal sampling will alias this variability". This is not correct. An undersampled time series of fluxes at 10 day intervals would still contain the signal of intermittent entrainment fluxes. The quantitative effect of that signal is included in the flux statistics if the undersampled time series is long enough. For example, the recent paper in Nature Geoscience by Johnson and Bif shows that the diel signal of primary productivity can be extracted quantitatively from 10 day profiling float data. The JGR:Oceans paper by Carranza et al. (<https://doi.org/10.1029/2018JC014416>) demonstrates that the intermittent effects of storm-driven mixing of plankton can be quantified. The effects of processes that are undersampled in space can also be resolved, as shown by the recent paper J. Mar. Systems paper by Su et al. (<https://doi.org/10.1016/j.jmarsys.2021.103569>), which shows the effects of eddies on plankton distributions.

Finally, one set of measurements over a two month period is probably not sufficient to support the statement (line 72) "we quantify the synoptic variability around the entire subpolar Southern Ocean".

In conclusion, this is an interesting and unique data set that gives insight into the mechanisms by which storms drive air-sea gas exchange. The statements as to its significance are overly broad, somewhat incorrect, and they detract from the impact.

Reviewer #2 (Remarks to the Author):

On this second round, I am satisfied that the authors have responded to my comments very well. The discussion of why the space scales of the fronts and synoptic Ekman flow is great, as well as that on the possible rectification to lower frequencies. My only remaining criticism is that the abstract should reflect some of this insight.

I support publication.

Reviewer #3 (Remarks to the Author):

The authors have done a very nice job of responding to significant, and constructive comments from all reviewers. In particular, the point of needing greater emphasis on biologically driven changes in DIC, and pCO₂. They have now included additional analyses of satellite-derived NPP and in situ estimates from optical sensors on the glider itself. I find that this strengthens their

analysis substantially.

With respect to my own comment about the lateral gradients in DIC and alkalinity, I appreciate the effort taken to include an expanded discussion of the magnitude and synoptic variation in these parameters, and included the alkalinity data in figure 1 in the main text.

I would now recommend that the manuscript is suitable for publication.

REVIEWERS' COMMENTS

Reviewer #1 (Remarks to the Author):

We thank the reviewer for their comments.

As I noted in my first review, this is an impressive data set. There is some nice analysis. Those results are marred by some overly broad conclusions. As an example, the abstract states “we show that the air-sea gradient of CO₂ is dominated by synoptic storm-driven ocean variability”. That is really not correct. As I noted in my first review, the air-sea gradient is dominated by the effects of biological productivity that create a negative air-sea flux. The storm-driven variability is modulating the flux, but it is not dominating the flux. If it were not for the overwhelming signal due to biological uptake of CO₂, the air-sea flux would be positive due to the effects of seasonal heating, as is the case further north. The authors recognize this, albeit indirectly, with the statement they have added (line 191 and further in the Rebuttal document), “These two events are associated with two of the largest CO₂ outgassing peaks observed during the experiment (Fig. 2e) and shifted the mean sea-to-air FCO₂ from -0.12 mol C m⁻² yr⁻¹ (i.e., the mean excluding these two events of positive FCO₂) to -0.06 mol C m⁻² yr⁻¹ (i.e., the mean of the full record in Fig. 2e including these two events).” It is only through biological uptake that the flux is set to a negative value. It is certainly not surprising that storms modulate the air-sea flux, but the storms do not dominate the flux. The authors are overstating the impact of a nice study that illustrates the mechanisms by which storms can affect the air-sea flux.

We have changed “dominated” to “modulated” in the abstract, see sentence below:

*Using an unprecedented multi-glider dataset combining air-sea CO₂ fluxes and ocean turbulence we show that the air-sea gradient of CO₂ is **modulated** by synoptic storm-driven ocean variability (20 μatm, 1-10 days) through two mechanisms.*

Another example of an overly broad statement (line 434) is “The probability of sampling short storm driven entrainment events like those observed by the multi-glider deployment with a 10 day (e.g., floats) or greater sampling period (e.g., ships) is very low and the response to entrainment is obscured by Ekman advection in any case. Thus, it is not currently possible to observationally quantify intermittent synoptic entrainment fluxes across the entire subpolar Southern Ocean as we do in the SE Atlantic with this data from paired gliders; coarser spatio-temporal sampling will alias this variability”. This is not correct. An undersampled time series of fluxes at 10 day intervals would still contain the signal of intermittent entrainment fluxes. The quantitative effect of that signal is included in the flux statistics if the undersampled time series is long enough. For example, the recent paper in Nature Geoscience by Johnson and Bif shows that the diel signal of primary productivity can be extracted quantitatively from 10 day profiling float data. The JGR:Oceans paper by Carranza et al. (<https://doi.org/10.1029/2018JC014416>) demonstrates that the intermittent effects of storm-driven mixing of plankton can be quantified. The effects of processes that are undersampled in space can also be resolved, as shown by the recent paper J. Mar. Systems paper by Su et al. (<https://doi.org/10.1016/j.jmarsys.2021.103569>), which shows the effects of eddies on plankton distributions.

The reviewer's point is well taken. We have changed “not currently possible” to “difficult” below excerpt as highlighted by the reviewer:

*“The probability of sampling short storm-driven entrainment events like those observed by the multi-glider deployment with a 10 day (e.g., floats) or greater sampling period (e.g., ships) is very low and the response to entrainment is obscured by Ekman advection in any case. Thus, it is **difficult** to observationally quantify intermittent synoptic entrainment fluxes across the entire subpolar Southern Ocean as we do in the SE Atlantic with this data from paired gliders; coarser spatio-temporal sampling **may** alias this variability^{17,28}. “*

Finally, one set of measurements over a two month period is probably not sufficient to support the statement (line 72) “we quantify the synoptic variability around the entire subpolar Southern Ocean”.

It is explained in the preceding half of the sentence that we apply a conceptual mixed-layer model that captures this variability. We have also used a Saildrone circumpolar dataset (~7 months sampling across multiple sites in the subpolar region) to interrogate the robustness of our estimate, Figure 4a.

We have added mention of the saildrone data, and furthermore we have changed the wording from “quantify” to “estimate”, see bold text below.

*... Then, using a conceptual ocean mixed layer model that captures the observed synoptic variability of $\Delta p\text{CO}_2$ in the observations, we **estimate** the synoptic variability around the entire subpolar Southern Ocean. **The robustness of this extrapolation is assessed with circumpolar observations measured by a Saildrone²⁸. Finally, we discuss the implications of synoptic processes for the mixed layer carbon budget and F_{CO_2} on longer timescales.***

We have also reworded the abstract to have a slight shift of emphasis from “quantitative” estimates to process modelling - see abstract changes in response to reviewer 2.

In conclusion, this is an interesting and unique data set that gives insight into the mechanisms by which storms drive air-sea gas exchange. The statements as to its significance are overly broad, somewhat incorrect, and they detract from the impact.

Reviewer #2 (Remarks to the Author):

On this second round, I am satisfied that the authors have responded to my comments very well. The discussion of why the space scales of the fronts and synoptic Ekman flow is great, as well as that on the possible rectification to lower frequencies. My only remaining criticism is that the abstract should reflect some of this insight.

I support publication.

We thank the reviewer for their constructive comments on our manuscript and are happy to hear they approve of our changes. Regarding the abstract, we are constrained by the tight word limit, however we have adapted it to read:

*The subpolar Southern Ocean is a critical region where CO₂ outgassing influences the global mean air-sea CO₂ flux (F_{CO_2}). However, the **processes** controlling the outgassing remain elusive. We show, using an unprecedented multi-glider dataset combining F_{CO_2} and ocean turbulence, that the air-sea gradient of CO₂ (ΔpCO_2) is modulated by synoptic storm-driven ocean variability (20 μatm , 1-10 days) through **two processes**. Ekman transport explains 60% of the variability, and entrainment drives strong episodic CO₂ outgassing events of 2-4 mol m⁻² yr⁻¹. Extrapolation across the subpolar Southern Ocean using a **process** model **shows how ocean fronts spatially modulate** synoptic variability in ΔpCO_2 (6 μatm^2 average) and **how spatial variations in stratification influence** synoptic entrainment of deeper carbon into the mixed layer (3.5 mol m⁻² yr⁻¹ average). These results not only constrain aliased-driven uncertainties in F_{CO_2} but also the effects of synoptic variability on slower seasonal or longer ocean physics-carbon dynamics.*

Reviewer #3 (Remarks to the Author):

The authors have done a very nice job of responding to significant, and constructive comments from all reviewers. In particular, the point of needing greater emphasis on biologically driven changes in DIC, and pCO₂. They have now included additional analyses of satellite-derived NPP and in situ estimates from optical sensors on the glider itself. I find that this strengthens their analysis substantially.

With respect to my own comment about the lateral gradients in DIC and alkalinity, I appreciate the effort taken to include an expanded discussion of the magnitude and synoptic variation in these parameters, and included the alkalinity data in figure 1 in the main text.

I would now recommend that the manuscript is suitable for publication.

We thank the reviewer for a supportive evaluation and constructive reviews in the previous round.